# Experimental demonstration of peripherally-excited antenna arrays

Ayman H. Dorrah [1] & George V. Eleftheriades[1✉]

Emerging technologies such as 5G communication systems, autonomous vehicles and satellite Internet have led to a renewed interest in 2D antennas that are capable of generating fixed/scannable pencil beams. Although traditional active phased arrays are technologically suitable for these applications, there are cases where other alternatives are more attractive, especially if they are simpler and less costly to design and fabricate. Recently, the concept of the Peripherally-Excited (PEX) antenna array has been proposed, promising a sizable reduction in the active-element count, especially when compared with traditional phased arrays. Albeit at the price of exhibiting some constraints on the possible beam-pointing directions. Here, we demonstrate the first practical implementation of the PEX antenna concept, and the proposed design is capable of generating single or multiple independently scannable pencil beams at broadside and tilted radiation directions, from a shared radiating aperture. The proposed structure is also easily scalable to higher millimeter-wave frequencies, and can be particularly useful in MIMO and duplex antenna applications, commonly encountered in automotive radars, among others.

---

[1] The Edward S. Rogers Sr. Department of Electrical and Computer Engineering, University of Toronto, Toronto, ON M5S 3G4, Canada.
✉email: gelefth@waves.utoronto.ca

The proliferation of new innovations such as 5G communication systems, autonomous vehicles and satellite Internet, has renewed the interest in highly directive 2D antennas to realize long-distanced point-to-point communication and/or object detection. However, these 2D antennas are typically operated at millimeter-wave frequencies which increases the cost of designing, fabricating, and deploying such antenna systems. For applications that require fixed-beam generation, parabolic reflector antennas are regarded as prime candidate solutions, owing to their directive beams and relatively wide operation bandwith. Nevertheless, their inherently large size and heavy weight hinder some applications. Accordingly, it becomes more suitable to deploy light-weight printed-circuit-board (PCB) and waveguide antennas as proposed in refs. [1–24]. Notably, the majority of real-life applications, mandating portable and mobile operation, crucially require antennas that are capable of generating not only highly directive but scannable pencil beams as well. From a technical standpoint, traditional active phased arrays can perfectly achieve scannable beams at the expense of high deployment cost.

The tradeoff between scannability and cost can be mitigated by deploying alternative 2D antennas that are simpler to design and less costly to fabricate than traditional phased arrays. For example, previous research efforts lead to the introduction of concepts such as thinned and sparse arrays[25–31]. These special phased arrays can reduce the number of active antenna elements required by leaving some antenna elements unexcited, or removing them completely from the grid, and in some extreme cases, breaking the periodicity of the 2D grid entirely. To design these lower-element-count phased arrays, various optimization algorithms are often employed. An alternative technique is to stack or interleave sub-antenna arrays as proposed in refs. [32–37]. Overall, these approaches and others fail to achieve a sufficient reduction in the active-element count. Furthermore, there is usually a cost to achieving a significant reduction of active elements using these techniques, such as a lower directivity, higher sidelobe levels, or a limited beam scanning range from broadside. In addition, some of these overlapped phased array techniques require complex feeding networks, which can be cumbersome[35–37].

An interesting alternative antenna concept to phased arrays is the "Continuous Transverse Stubs (CTS) Array"[38]. This concept does not require phase shifters, instead, it typically relies on some relative mechanical rotation between its constituent components[39]. The CTS array usually comprises a parallel-plate waveguide perforated at the top with a 1D array of long slots, which are excited with a plane wave orthogonal to the long side of the slots. The slots radiate a pencil beam in free space through a traveling leaky-wave antenna (LWA) operation (series-fed operation). The generated pencil beam can be scanned in both elevation and azimuth along predefined contours, by mechanically changing the relative orientation between the incident plane wave and the 1D slots. This CTS array concept has been realized using a fully-metallic structure with three separate metal layers in ref. [39]. These metal layers are implemented using contact-less gap waveguides, allowing the mechanical rotation of the top radiating layer relative to the bottom feeding layers, thus changing the relative orientation between the excited plane wave and the long slots. This leads to a continuous scanning of the generated pencil beam in both azimuth and elevation along a single predefined contour. Note that the long-slot radiating unit cell cannot achieve successful radiation when excited with a plane-wave along its long edge, which limits the possible scanned contours from the structure. To achieve full-space scanning, it is required to mechanically rotate the three metal layers together, in addition to an independent relative rotation between them. It is worth highlighting that in addition to the traveling-wave (series-fed)

operation in ref. [39], it is possible to operate the CTS array in a parallel-fed fashion, which offers excellent H-plane scanning and very large operational bandwidths[40]. This parallel CTS antenna solution is used extensively in SatComm-on-the-move commercial systems, even though mechanical rotation is still needed to achieve full-space scanning.

On the other hand, numerous switched-beam antenna designs have been proposed[41–44], that are operated in a similar manner to the CTS array, and are also able to combat the aforementioned tradeoff between scannability and cost. These antenna designs also eliminate the need for phase shifters, and are able to change the relative orientation between the excited plane wave and the radiating structure by using single-layer[41,42] or multi-layer[43,44] switched multi-port beamforming networks (lenses). These networks exhibit multiple switched-input ports situated along the focal plane/region of a lens such as a parabolic or Luneburg lens, and they leverage the Fourier-transform properties of the lens to excite a LWA structure with switched plane waves, by activating the different input ports. Thus, this LWA structure is operated using a switched-beam operation, and each input port corresponds to a discrete pencil beam in space situated along predefined contours. One difficulty with these switched-beam antennas is that they cannot scan the generated pencil beams continuously, which is the price of the switched-beam operation, and usually the 3-dB beamwidth of the generated beams needs to be carefully optimized for an overlapping spatial coverage. This becomes more challenging when very narrow beams are required. Another potential drawback is that some lens designs such as parabolic reflectors are inherently constrained to reduced spatial scanning (reduced angular coverage), due to eventual aberrations and spillover loss from feed elements displaced farther from the focal point. Also, these switched-beam antennas are typically not designed for broadside operation, which is inconvenient in some applications. More importantly, the LWA unit cells in refs. [41–44] are only excited along a single side of the LWA, and are not excited from the orthogonal side (although it is possible in principle), which limits the number of possible scan contours from these switched-beam antennas. Thus, a LWA unit cell design that is optimized for broadside operation, while allowing excitation from the two orthogonal sides of the LWA is highly desirable, and will increase the number of possible scan contours from the LWA, effectively constituting multiple antennas in one shared radiating aperture.

In a nutshell, the aforementioned CTS and switched-beam antennas either require mechanical rotation or input-port switching to change the relative orientation between the excited plane wave and the slot array. Recently, a remarkable phased-array alternative concept—the Peripherally-Excited (PEX) Antenna Array[45–48]—has been proposed, and it scans the excited plane wave inside the parallel-plate waveguide in a completely different manner. The PEX antenna concept stems from the Huygens' Box structure[49–53], which enables the excitation of plane waves inside a closed cavity along arbitrary directions. In the PEX antenna concept, the Huygens' box is made leaky by appropriately perforating its top plate, and exciting it with an underlying electronically-steered plane wave. These plane waves are generated by peripheral Huygens' sources situated along the edges (sides) of the radiating aperture, according to the Huygens' equivalence principle. The Huygens' box concept has been experimentally validated previously[49–52], where it has been demonstrated that plane waves can be excited along arbitrary directions inside a fully-closed metallic cavity, using peripherally-placed Huygens' sources. This is a quite remarkable feat as metallic cavities only inherently support standing waves, and the PEX array concept leverages these traveling plane waves and uses them to excite a radiating leaky-wave structure, generating electronically-steerable pencil beams.

The PEX array achieves a sizable reduction in the active-element count compared to traditional phased arrays, especially for larger aperture sizes[45,46], as it requires active antenna elements solely along the edges (sides) of the PEX cavity. The radiating aperture itself is merely filled by a periodic arrangement of passive antenna elements, and the structure is operated as a LWA. This significantly reduces the active-element count compared to conventional phased arrays since the active-element count now scales with the perimeter of the radiating aperture, instead of its area. This reduction in the active-element count becomes more pronounced as the size of the radiating aperture gets larger, i.e. when high directivity pencil beams are required. In addition, the proposed PEX unit cell has been designed and optimized with 2D plane-wave excitation in mind, hence, it can be operated from all its sides, leading to more scanned planes. The proposed design can also be leveraged to generate multiple pencil beams from the PEX array simultaneously, thus mimicking the role of multiple independent antennas sharing the same radiating aperture, which saves valuable real-estate. The implications of this antenna are far-reaching as it can be potentially deployed in multiple-beam (MIMO) and duplex applications with simultaneous transmit and receive operation from the same antenna (not necessarily from the same direction).

This paper herein introduces the first experimental demonstration of the PEX antenna concept, while highlighting several desirable features and capabilities of the proposed antenna concept. This paper tackles many of the implementation challenges for realizing the PEX array, namely, implementing the PEX array using a low-cost and low-profile manufacturing process such as printed-circuit boards (PCBs), designing appropriate active (Huygens') sources to be used as peripheral sources inside these PCBs, introducing suitable electronic phase-shifting PCB sub-systems, suppressing the mutual coupling between the closely-spaced peripheral sources, and designing a radiating unit cell that is capable of radiating at broadside and tilted angles successfully without exhibiting any open bandgaps, among other challenges. Our proposed antenna can be easily scaled to higher millimeter-wave frequencies, as discussed later. In the proposed implementation, a Huygens' source is constructed from a coaxial feed backed by an array of metallic vias, which is entirely compatible with standard PCB fabrication, and can be easily embedded inside commercial two-layer dielectric substrates. Furthermore, the proposed PEX unit cell is capable of generating electronically-steered pencil beams at broadside and tilted radiation directions, by adopting techniques in closing the bandgap for broadside leaky-wave antennas[54,55]. In the following "Results" section, we describe the PEX antenna array concept and its relation to the Huygens' box. Additionally, we propose a practical peripheral Huygens' source implementation. Then, we present a specially-engineered radiating unit cell design that is capable of achieving successful radiation at broadside and tilted angles. After that, we show a practical implementation of the PEX antenna design using that unit cell, and the resulting structure is very easily scalable to higher millimeter-wave frequencies. We fabricate and experimentally test a prototype of the antenna design and demonstrate its versatility and practicality. In the "Discussion" section, we conclude the paper with some observations and remarks, including how to extend the scan range of the PEX array, beyond its predefined scanned-planes, by a mechanical rotation or employing tunable substrates. In the "Methods" section, we describe the main methods and procedures of designing, simulating, fabricating and testing the proposed PEX antenna array. Furthermore, we provide additional information pertaining to the phase-shifting feeding network and the frequency response of the PEX array in the Supplementary Notes.

## Results

### Schelkunoff's equivalence principle

The peripherally-excited (PEX) antenna array stems from the Huygens/Schelkunoff equivalence principle. In simple terms, the equivalence principle states that the electromagnetic wave in a given region (Volume) is unique and is solely determined by the tangential electric and magnetic fields, and the electric and magnetic surface currents along the boundary surface enclosing that region[56,57]. A particular case of interest here is the two-dimensional (2D) variation of Schelkunoff's equivalence principle (see Fig. 1a), where electromagnetic waves $\overline{E}$-$\overline{H}$ are defined inside a surface $S_i$ that is outline by a closed contour $C$. External to this contour is a surface $S_o$, that is free from any electromagnetic waves, and is conveniently filled with a perfect-electric conductor (PEC). To realize the discontinuous fields across the contour boundary, only a magnetic surface current $\overline{M}_s$ needs to be impressed along the contour $C$. The PEC region negates the need for any electric surface currents $\overline{J}_s$ along the contour $C$, as $\overline{J}_s$ is shorted by the PEC. The contour $C$ can exhibit any outline, shape and size, and it maybe polygonal or curvilinear. For a vertically polarized electromagnetic wave, this setup can be realized in practice using two closely-spaced PEC planes (a parallel plate waveguide with subwavelength thickness). Then, the contour $C$ defines where the PEC side walls are placed, along which a magnetic surface current $\overline{M}_s$ is impressed.

### Huygens' Box

The resulting Huygens' Box structure is a thin flat metallic cavity with PEC side walls that are lined with effective magnetic surface currents (effective Huygens' sources) (see Fig. 1b)[45–53]. The magnetic surface currents can be effectively realized using an array of electric current sources that are separated by a distance $p$, placed along the edges (sides) of the metallic cavity. The separation distance $p$ is ideally chosen to be smaller than or equal to $\lambda/2$, to satisfy sampling theorem requirements[45–53], where $\lambda$ is the wavelength inside the metallic cavity. This allows the Huygens' box to excite any electromagnetic wave distribution within the cavity, as long as the electromagnetic wave exhibits a vertically polarized electric field and is a solution to Maxwell's equations, such as a single or multiple propagating plane waves[45–53], even though plane waves do not belong to the inherent modal set of typical metallic cavities.

### PEX antenna array

The PEX antenna array is a Huygens' box with perforations or slots (passive antenna elements) on the top side of the cavity (see Fig. 1c). The perforations allow the excited plane wave(s) underneath to radiate one or more pencil beams to free space, similar to traveling-wave (leaky-wave) antennas. The PEX array can be realized using a double-sided printed-circuit board (PCB) that forms the top and bottom metallic plates, and an array of metallic vias that realizes the metallic side walls. In turn, the magnetic surface currents can be implemented using coaxial ports as proposed in Fig. 1d. Each coaxial feed injects a circulating electric current $I_e$ along the metallic side walls, which realizes an effective magnetic surface current $\overline{M}_s$ as required. The close placement of these coaxial feeds to the metallic side walls constitutes a major challenge, as PECs tend to cancel any tangential electric currents placed substantially close to them. As a result, the feed can undergo a significant impedance mismatch which compromises its ability to inject power into the PEX cavity. Hence, the distance between the coaxial feeds and the metallic side walls $d_p$ is carefully chosen, and is set to more than a quarter of the wavelength ($\lambda/4$), making the current loop formed slightly smaller than the wavelength ($\lambda$). This allows the "scattering" from the electric currents on the metallic side walls to be in phase with that of the coaxial feed, preventing any impedance mismatch

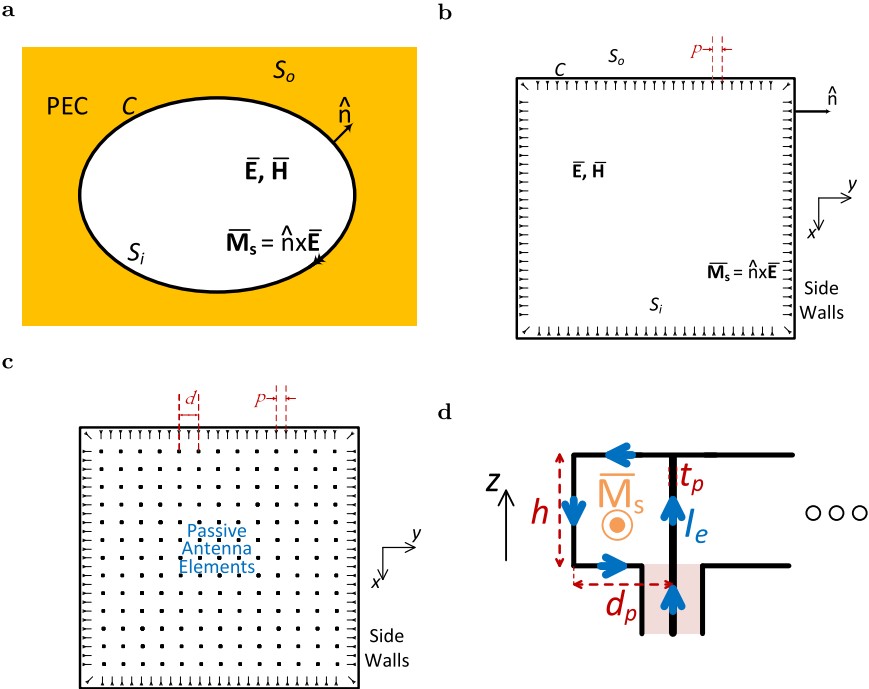

**Fig. 1 PEX array concept. a** A general depiction of Schelkunoff's equivalence principle with magnetic surface currents lining the periphery of a region surrounded by a perfect-electric conductor (PEC). The electromagnetic fields inside the region are solely determined from the tangential magnetic surface currents on the Contour $C$. **b** A general illustration of the Huygens' Box, which is a practical implementation of Schelkunoff's equivalence principle using a metallic cavity. **c** A general schematic depiction of the PEX antenna array, which is essentially a Huygens' Box with additional passive antenna elements (slots) on the top side, allowing the plane waves underneath to radiate. **d** Side view of the proposed peripheral PEC-backed effective magnetic current sources which inject a circulating electric current into the side walls of the metallic cavity. They are created using a coaxial feed, and are used to realize the peripheral Huygens' sources required for the PEX cavity. (Dimensions: $p = 7.15$ mm, $d = 14.3$ mm, $\epsilon_r = 2.2$, $d_p = 5.5$ mm, $t_p = 1.4$ mm, and $h = 1.575$ mm on a Rogers RT/duroid 5880 1oz substrate).

problems, which facilitates the injection of power into the PEX cavity. More details on the reflection, coupling and transmission response between the various sides of the PEX array are discussed in Supplementary Note 2.

**Accidental degeneracy**. The PEX antenna array requires periodic perforations (passive antenna elements) on the top side of the PEX cavity to allow the plane wave(s) underneath to radiate pencil beam(s) to free space. The perforations ideally should enable radiation along broadside and other tilted directions in a traveling-wave (leaky-wave) manner. However, for such periodic structures, an open bandgap commonly emerges between the relevant 3–4 eigenmodes in the dispersion relation[23,54,55,58], and results in a frequency range in the dispersion relation that is not covered by any eigenmode solution. Whenever a wave is excited in the frequency range of the open bandgap, it is not allowed to propagate inside the structure, and is entirely reflected back towards the source of excitation. Such an open bandgap typically exists at the $\Gamma$-point in the Brillouin diagram, which corresponds to the broadside radiation direction. Hence, successful radiation at and around broadside is a quite challenging endeavor. To overcome this challenge, the bandgap can be closed by achieving accidental degeneracy between the eigenmodes at the $\Gamma$-point in the dispersion relation[23,54,55,58]. This is achieved by manipulating the eigenmodes of the periodic structure, and forcing them to coexist at the $\Gamma$-point at exactly the same resonance frequency. For a traveling-wave antenna, the eigenmodes are also required to have a balanced radiation strength (similar $Q$-factors) to guarantee the accidental degeneracy of the eigenmodes, and the complete closure of the bandgap at the $\Gamma$-point[23,54,55]. Note that the $Q$-factor in this case is directly related to the leakage constant

($\alpha$) of the unit cell, where a lower $Q$-factor, corresponds to a higher leakage constant, and more radiation from the individual unit cells.

**Radiating unit cell with closed bandgap**. The radiation from the proposed unit cell is achieved by cross-shaped slots (see Fig. 2a), and the dimensions of these slots control the amount of leakage constant and the $Q$-factor. These cross slots have been chosen for their immunity against the typical scan blindness of other slot shapes, where scan blindness in this context refers to the inability of some slots to achieve significant radiation when the traveling wave is oriented in a specific way relative to the slot. As a result, these slots achieve successful radiation at broadside and tilted directions while maintaining a high level of polarization purity for the generated pencil beam(s) (i.e., low cross-polarization levels). More importantly, four semi-square shaped slots are included to realize accidental degeneracy of the eigenmodes at the $\Gamma$-point, and close the bandgap in the dispersion relation. This is confirmed from the 2D dispersion relation depicted in Fig. 2b, and the 2D dispersion contours depicted in Fig. 2c, which both show that all the eigenmodes are coexistent at the $\Gamma$-point. This clearly demonstrates the successful closure of the bandgap at 13.1 GHz, and suggests that broadside radiation is possible using this specially engineered unit cell. For the proof-of-concept demonstration in this paper, the frequency of operation (13.1 GHz) has been chosen for demonstration purposes. It is a good compromise as it is high enough for the resulting 2D array design shown later to be physically small and manageable, while still being low enough to allow the use of inexpensive commercial connectors, cables, and terminations. The corresponding electric field distribution of the individual eigenmodes at the $\Gamma$-point (13.1 GHz) are shown in

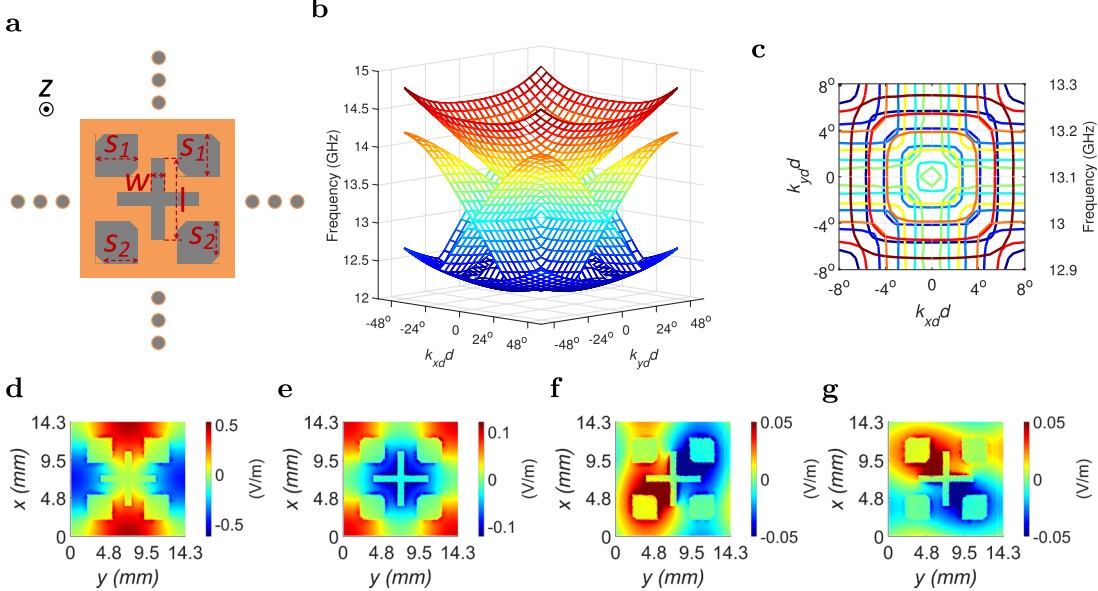

**Fig. 2 Unit cell simulation.** The specially engineered PEX array unit cell with a closed bandgap in the dispersion relation: **a** Top view of the unit cell showing the cross-shaped radiating slots, and the semi-square slots that are added to achieve accidental degeneracy between the eigenmodes, and close the bandgap at the Γ-point. **b, c** The corresponding full-wave simulated 2D dispersion relation and 2D dispersion contours showing a closed bandgap at the Γ-point. **d–g** The corresponding full-wave simulated electric field distribution (V/m) of the 1st, 2nd, 3rd, and 4th eigenmodes at the Γ-point, respectively. (Dimensions: $w = 1$ mm, $l = 7$ mm, $S_1 = 3.3$ mm, $S_2 = 2.7$ mm and unit cell size = 14.3 mm × 14.3 mm × 1.575 mm).

Fig. 2d–g. Noticeably, the electric field distributions around the cross-shaped slots are antisymmetric, especially for the 3rd and 4th eigenmodes, making the unit cell capable of achieving successful radiation (leakage) from the underlying electric field. More details about the process of designing and optimizing the unit cell, and closing the bandgap are discussed in the "Methods" section.

**Analytical beam-pointing directions.** To construct the PEX antenna array, the unit cells are placed in a 2D periodic square lattice surrounded by the active peripheral sources (see Fig. 1c). The passive unit cells are individually excited by arbitrary propagating plane waves generated from the peripheral sources and having the general wave vector ($k_{xd}$, $k_{yd}$), which provides the individual unit cells with appropriate excitations (with the correct weight and/or phase). The expected radiation direction from the unit cells can be simply calculated by phase matching the plane wave propagating underneath the radiating slots ($k_{xd}$, $k_{yd}$) with the radiated free-space wave, leading to the following expressions (see[45,59] for the full derivation):

$$\sqrt{\epsilon_e}\cos(\psi) = \sin(\theta)\cos(\phi) - \frac{\lambda_o}{d}n, \qquad (1a)$$

$$\sqrt{\epsilon_e}\sin(\psi) = \sin(\theta)\sin(\phi) - \frac{\lambda_o}{d}m, \qquad (1b)$$

where $\lambda_o$ is the free-space wavelength, $\epsilon_e$ is the effective relative permittivity of the unit cell, $\psi = \tan^{-1}(k_{yd}/k_{xd})$ is the azimuthal orientation of the plane wave underneath the radiating slots measured from the $x$-axis, ($n$, $m$) are integer constants accounting for the possible Floquet modes that can be radiated from the structure, $\phi$ is the generated pencil beam azimuthal orientation measured with-respect-to the $x$-axis, and $\theta$ is the generated pencil beam tilted direction measured with-respect-to the $z$-axis (More details on the Theta-Phi ($\theta$-$\phi$) spherical coordinate system used are provided in Supplementary Note 4). These phase matching equations assume an infinitely periodic 2D array of unit cells, that can support the excitation of Floquet modes, however, the equations remain accurate for relatively large radiating apertures.

On the other hand, it is also possible to express the inverse equations that calculate the pencil-beam direction ($\phi$, $\theta$) as a function of the direction of plane wave ($\psi$) inside the PEX cavity[45,59]:

$$\sin^2(\theta) = \epsilon_e + \frac{\lambda_o^2}{d^2}(n^2 + m^2) + 2\sqrt{\epsilon_e}\frac{\lambda_o}{d}\left[n\cos(\psi) + m\sin(\psi)\right], \qquad (2a)$$

$$\tan(\phi) = \frac{\sqrt{\epsilon_e}\sin(\psi) + \frac{\lambda_o}{d}m}{\sqrt{\epsilon_e}\cos(\psi) + \frac{\lambda_o}{d}n}. \qquad (2b)$$

Note that for the preliminary structure shown in ref. [45], the unit cells comprise subwavelength square-shaped slots, which do not disturb the dispersion relation of the unit cells, compared to a parallel plate waveguide. Hence, the effective relative permittivity of the unit cells therein is merely that of the underlying dielectric medium ($\epsilon_r$). However, for the proposed PEX unit cell in Fig. 2a, the perturbations to the unit cells are significant, and the dispersion relation is quite different from that of an unperturbed parallel plate waveguide. Thus, an effective relative permittivity ($\epsilon_e$) is required to be defined, and it can be approximated from ($\epsilon_e = \lambda_o^2/\lambda_{PEX}^2$), where $\lambda_{PEX}$ is the wavelength within the PEX cavity. Thus, this means that the effective relative permittivity ($\epsilon_e$) depends on the relative permittivity of the dielectric filling the PEX cavity ($\epsilon_r$), the frequency of operation, and the size and shape of the slot arrangement used (see Fig. 2a). If the unit cell is assumed to be operated around the Γ-point of the dispersion relation (13.1 GHz), $\lambda_{PEX}$ is approximately equal to the periodicity of the unit cells (the size of the unit cells $d$), and the effective relative permittivity ($\epsilon_e$) can be estimated from $\epsilon_e = \lambda_o^2/d^2$. Using this value for $\epsilon_e$, the previous expressions simplify to:

$$\sin^2(\theta) = \frac{\lambda_o^2}{d^2}\left[1 + n^2 + m^2 + 2n\cos(\psi) + 2m\sin(\psi)\right], \qquad (3a)$$

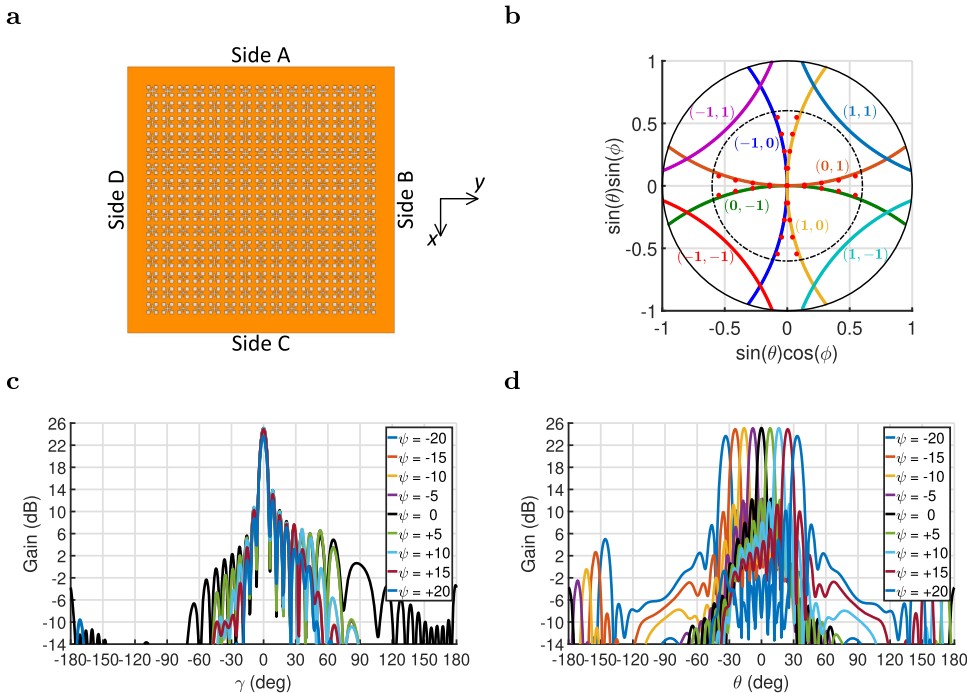

**Fig. 3 PEX array simulation.** Full-wave simulation of the 15 × 15 PEX antenna array: **a** Top view of the PEX array which is etched on a Rogers RT/duroid 5880 1oz substrate, showing the different sides of the array A, B, C and D, and the coordinate system used. (Dimensions = 245.8 mm × 245.8 mm × 1.575 mm). **b** Comparison between the analytical (solid) and the full-wave simulated (red dots) pencil-beam directions ($\phi$, $\theta$) at 13.1 GHz for multiple plane wave directions ($\psi$) exciting the PEX array, both showing a similar scanning behavior along predefined contours with a versatile range of tilted angles. **c, d** The full-wave simulated realized gain (dB) patterns plotted along the E-plane and H-plane, respectively, at 13.1 GHz, for various values of ($\psi$), when only side A is excited. Successful electronic scanning of the generated pencil beams is observed.

$$\tan\left(\phi\right) = \frac{\sin(\psi) + m}{\cos(\psi) + n}. \tag{3b}$$

The estimated pencil-beam directions ($\phi$, $\theta$) that are possible for the unit cell depicted in Fig. 2a are plotted later in Fig. 3b, considering the different Floquet modes (0, ±1), (±1, 0) and (±1, ±1) that are within the visible radiation region. To obtain these beam-pointing angles in Fig. 3b, all the possible plane wave directions ($\psi$) have been considered ($\psi \in [0, 2\pi]$). The corresponding grating-lobe limit for the proposed PEX unit cell is also plotted in Fig. 3b using a black dotted circle. Inside this circle, all the shown beam-pointing directions are realizable without the generation of any grating lobes, as they satisfy the unimodal condition. Note that because of the symmetric nature of the proposed unit cell, the generated pencil beams can possible scan along multiple orthogonal predefined contours as demonstrated by the lines inside the grating-lobe limit circle. On the other hand, the empty portions of the visible region in Fig. 3b correspond to directions where the generated beam cannot be pointed (at the Γ-point frequency). Thus, for extremely large antennas with very narrow beams, the portion of the visible region that can be practically covered by the generated beams will be reduced, because of the narrower beamwidth. Notably, the possible radiation beam-pointing directions ($\phi$, $\theta$) are constrained, as the generated pencil beams are scanned only along predefined contours, nevertheless, the scanning is considered acceptable especially the achieved range of tilted angles.

**PEX array simulation.** Full-wave simulations of a 15 × 15 PEX array are performed using the structure as shown in Fig. 3a, demonstrating the capabilities of the proposed PEX antenna (details

on the full-wave simulation setup are discussed in the "Methods" section). If the ports along a single side of the PEX array are all excited with an equal amplitude and a progressive phase shift, a single linearly polarized pencil beam is generated, with polarization along the propagation direction of the excited plane wave underneath the radiating slots. The generated pencil beam can be electronically scanned towards broadside and other tilted angles, by simply controlling the amount of the applied progressive phase shift, i.e., the plane wave propagation direction ($\psi$). Based on this concept, the full-wave simulated beam-pointed directions achieved by the PEX array at the center frequency of 13.1 GHz are shown in Fig. 3b, showing great agreement with the analytical expressions in Eq. (3). The center of this plot corresponds to broadside radiation, which is achieved when the ports along one side (such as side A) are excited in phase, and the remaining sides B, C and D are left unexcited. This excites a plane wave along the x-axis direction ($\psi = 0$), generating a broadside pencil beam as depicted in Fig. 3c, d. In these plots, the E-plane contains the direction of the peak of the beam and the electric field, whereas the H-plane contains the direction of the peak of the beam and the magnetic field (More details in Supplementary Note 4). For the radiation patterns plotted along the E-plane in Fig. 3c, an angle ($\gamma$) is defined as shown in Supplementary Note 4, with zero along the direction of the peak of the pencil beam. The generated broadside pencil beam achieves a realized gain of 25.1 dB, with an aperture efficiency of 84.6%, quite high for a traveling wave antenna. Notably, aperture efficiency compares the directivity of the beam to the maximum directivity achieved by a uniformly radiating aperture with the same physical size[56], thus, antennas that achieve such high aperture efficiencies guarantee the generation of narrow beams from a compact antenna that does not require much real estate. Such antenna designs are highly desirable in various applications such as satellite communications, point-to-point communications, and autonomous vehicles. On the other hand, the radiation efficiency of

**Table 1 Antenna parameters simulation. The full-wave simulated antenna parameters for a plane wave excited at various directions ($\psi$) inside the PEX array.**

| $\psi$ | −20° | −15° | −10° | −5° | 0° | +5° | +10° | +15° | +20° |
|---|---|---|---|---|---|---|---|---|---|
| $\phi$ (°) | 82 | 84 | 85.5 | 87.5 | Broadside | 92.5 | 94.5 | 96 | 98 |
| $\theta$ (°) | −33.5 | −24.5 | −16 | −8 | Broadside | 8 | 16 | 24.5 | 33.5 |
| R. Gain (dB) | 23.7 | 25.0 | 25.1 | 25.1 | 25.1 | 25.1 | 25.1 | 25.0 | 23.7 |
| 1 dB BW (%) | 2.6 | 2.5 | 2.4 | 2.5 | 2.5 | 2.5 | 2.4 | 2.5 | 2.6 |
| 3 dB BW (%) | 4.2 | 4.2 | 4.4 | 4.4 | 4.5 | 4.4 | 4.4 | 4.2 | 4.2 |
| Rad. Eff. (%) | 79.2 | 80.0 | 79.6 | 78.0 | 77.2 | 77.9 | 79.5 | 79.9 | 79.0 |
| Ap. Eff. (%) | 59.9 | 71.9 | 76.5 | 80.8 | 84.6 | 80.9 | 76.5 | 72.0 | 60.1 |
| Side excited | A | A | A | A | A | A | A | A | A |
| Power to side A (%) | 5.8 | 4.3 | 6.3 | 7.8 | 8.0 | 7.8 | 6.5 | 4.4 | 6.0 |
| Power to side B (%) | 4.5 | 2.6 | 1.8 | 1.1 | 1.3 | 2.1 | 3.0 | 1.4 | 0.8 |
| Power to side C (%) | 34.3 | 37.3 | 38.1 | 42.2 | 44.4 | 42.2 | 38.3 | 37.5 | 34.6 |
| Power to side D (%) | 1.2 | 1.7 | 3.3 | 2.1 | 1.3 | 1.1 | 1.7 | 2.6 | 4.4 |
| E-plane | | | | | | | | | |
| X-Pol (dB) | 17.1 | 19.1 | 20.5 | 23.6 | 55.4 | 23.5 | 20.4 | 19.0 | 17.0 |
| SLL (dB) | 11.7 | 11.8 | 11.2 | 11.3 | 11.2 | 11.3 | 11.2 | 11.8 | 11.7 |
| H-plane | | | | | | | | | |
| X-Pol (dB) | 18.0 | 19.7 | 21.4 | 24.5 | 30.9 | 24.3 | 21.3 | 19.6 | 18.0 |
| SLL (dB) | 13.4 | 13.6 | 13.2 | 12.8 | 12.8 | 12.9 | 13.3 | 13.7 | 13.5 |

the design is 77.2% where the remaining power is lost to heat due to the metallic and dielectric losses. Overall, the simulation results confirm that the proposed peripheral excitations are well matched to the PEX cavity, and do not suffer from any current cancellation or mutual coupling problems.

From the total power excited into side A, 8.0 is reflected back to side A, 1.3% is coupled to side B, 1.3% is coupled to side D, and 44.4% is transmitted across the PEX array to side C. Needless to say that a bigger PEX array will exhibit less power lost to the resistive terminations along side C, albeit at the price of an accompanying reduction in the aperture efficiency achieved, as the PEX antenna is constructed from a uniform leaky-wave antenna. As a result, the proposed PEX antenna array experiences an inevitable tradeoff between aperture efficiency, radiation efficiency, and termination efficiency. The latter refers to any remaining unradiated power that is lost to the resistive terminations. To mitigate this tradeoff, it is possible to design a PEX antenna array with tapered radiation from the unit cells, which is a standard practice for leaky-wave antennas that aim at achieving high aperture and radiation efficiencies simultaneously.

Additionally, tilted pencil beams are generated by exciting one side of the PEX array (such as side A) with a progressive phase shift, which excites a plane wave at a tilted angle with respect to the x-axis ($\psi \neq 0$). Sample tilted pencil beams are generated as shown in Fig. 3c, d, from plane waves excited along ($\psi = \pm5°$, ±10°, ±15°, and ±20°), respectively. Note that although the PEX array is designed for continuous beam steering by tuning the peripheral feeds, the performance is characterized for only a discrete set of descriptive cases for simplicity. The electric-field polarization of the generated beams is again along the propagation direction the excited plane wave, i.e., approximately along the x-axis when only side A is excited, and the corresponding orientations of the E-plane and H-plane are as described in Supplementary Note 4. Notably, it is sufficient to solely excite side A for these directions of plane wave propagation ($\psi$), as the corresponding plane waves form a considerably small angle with the x-axis, and there is no need to invoke any excitations from sides B, C, or D. The antenna parameters of all these pencil beams are summarized in Table 1. It is observed that the generated pencil beams achieve acceptable levels of realized gain, X-pol, SLL, radiation and aperture efficiencies. Also, a breakdown of the different portions of power that reach the four sides of the PEX

array when side A is excited is included in Table 1. This power breakdown does not include the effects of the feeding network that will be later used to excite the PEX array in the experiment. For the different cases, it is observed that a large portion of the incident power at side A reaches the opposite side C. Again, a larger PEX array will exhibit more radiation from the leaky wave, and less power will reach side C as a result, however, the aperture efficiency will inevitably drop as well (if the radiation is not tapered), as described earlier.

Notably, the proposed PEX antenna array achieves radiation by exciting a traveling leaky-wave mode. Hence, the generated pencil beams undergo a slight spatial scanning as the frequency of operation is changed, as demonstrated in more detail in Supplementary Note 2. The 1 and 3 dB bandwidths achieved by the generated pencil beams at the center frequency (13.1 GHz) for different plane-wave directions ($\psi$) are also shown in Table 1. For instance, at broadside, the generated pencil beam achieves a 2.5% 1 dB-bandwidth and a 4.5% 3 dB-bandwidth. This instantaneous bandwidth constitutes the actual useful bandwidth that can be used for communications, as it represents the frequency bandwidth where the beam scanning with frequency leads to a 1 or 3 dB attenuation. Note that this instantaneous bandwidth is limited by the spatial scanning of the generated pencil beams, and not beam degradation. The proposed PEX array can in principle be operated over a wider bandwidth while maintaining a directive beam, without beam degradation (as shown in Supplementary Note 2).

Notably, from the symmetry of the proposed PEX array, the additional possible beam-pointing directions, realized by exciting the other sides B, C, and D, can be inferred by analyzing the representative examples in Table 1, while cross-referencing with Eq. (3) and Fig. 3b.

**Multiple-beam simulation.** The principle of superposition can be leveraged by the PEX antenna array where more than one plane wave are generated simultaneously inside the PEX cavity. This generates multiple pencil beams at broadside and/or tilted angles. For example, side A is excited by a plane wave at ($\psi = 0°$), and side B is excited by a plane wave at ($\psi = 285°$), which radiates two pencil beams as depicted in Fig. 4a, d, where one beam is pointed towards broadside, and the other beam towards a tilted angle. It is

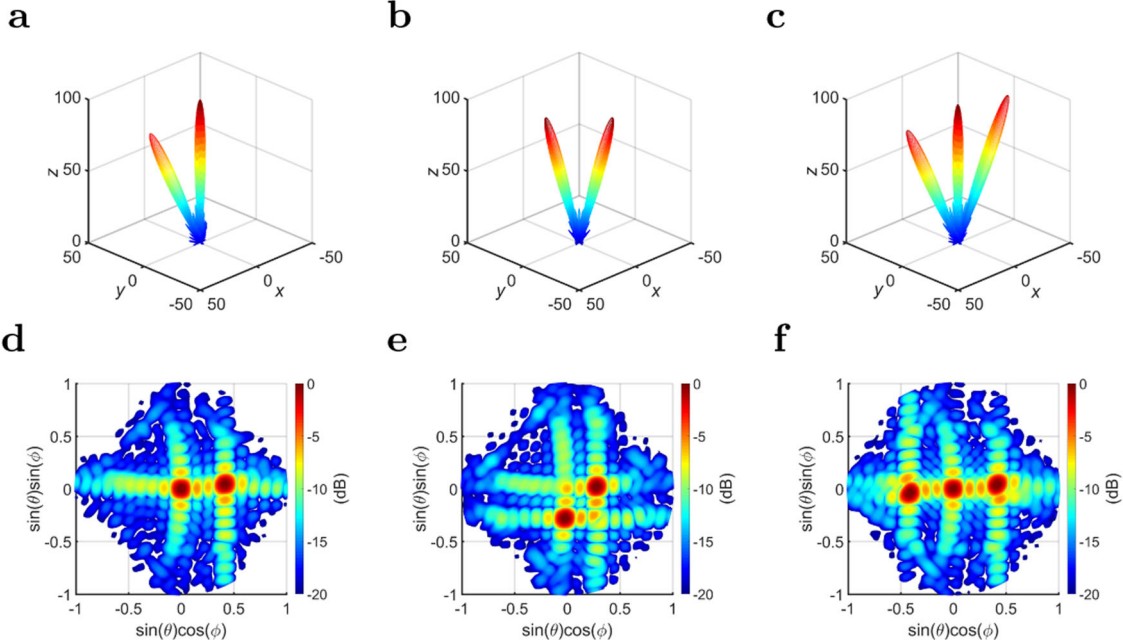

**Fig. 4 Multiple-beam simulation.** The normalized full-wave simulated radiation patterns for various values of ($\psi$), when both sides A and B (and C) of the 15 × 15 PEX array are excited simultaneously. **a**–**c** The full-wave simulated far-field 3D gain patterns for the cases: **a** $\psi = 0°$ and 285°, **b** $\psi = -10°$ and 280°, and **c** $\psi = 0°$, 285°, and 105°, plotted in linear scale (V/m) and normalized to an arbitrary value of 100. **d**–**f** The full-wave simulated far-field U-V gain patterns for the cases: **d** $\psi = 0°$ and 285°, **e** $\psi = -10°$ and 280°, and **f** $\psi = 0°$, 285°, and 105°, plotted in dB scale. All the beams can be independently scanned, and a multitude of radiation directions are achieved.

also possible to generate two beams both pointing at tilted angles simultaneously. For instance, side A generates a plane wave at ($\psi = -10°$), and side B generates a plane wave at ($\psi = 280°$), which radiate two tilted pencil beams as depicted in Fig. 4b, e. Furthermore, superposition can be applied even further where three or more pencil beams are generated. As a demonstration, side A generates a plane wave at ($\psi = 0°$), side B generates a plane wave at ($\psi = 285°$), and side C generates a plane wave at ($\psi = 105°$), which radiate three pencil beams as depicted in Fig. 4c, f. All of the multiple beams presented in Fig. 4 exhibit similar antenna parameters to those shown in Table 1.

Note that one of the main challenges in achieving multiple-beam or duplex operation from any antenna is the mutual coupling that may occur between the different beams, which limits the radiation efficiency of the antenna and prevents the successful duplex operation[60]. To minimize this unwanted mutual coupling, it is typically required that the transmitted and received beams exhibit a low beam-coupling factor, i.e., the beams are required to be orthogonal[60]. The proposed PEX antenna achieves this by exhibiting orthogonal polarizations for the beams generated from sides A and B, as the electric-field polarization of the radiated beams is along the propagation direction of the excited plane wave underneath the radiating slots (which is orthogonal for sides A and B). Thus, this causes a remarkably low mutual coupling between the orthogonal sides of the PEX array, and there are no coupling or degradation effects between the generated beams by sides A and B (see Table 1 and Supplementary Note 2), which enables the successful multiple-beam or duplex operation of the proposed antenna array. Thus, the proposed PEX array is capable of generating single and/or multiple pencil beams, that are independently scanned along predefined contours with a versatile range of tilted angles.

More importantly, the results presented here correspond to a 15 × 15 PEX array, with 31 ports along each of the four sides of the array. For single-beam generation, only a single side needs to be excited for most of the radiation directions, which sufficiently

excites the 225 radiating unit cells of the PEX array. Hence, the proposed PEX achieves a sizable reduction in the active-element count, albeit for the cost of providing limited beam-point directions. The reduction in active-element count is more remarkable for larger PEX array sizes[45]. On the other hand, it is worth highlighting that the proposed PEX antenna array is a traveling-wave antenna, thus, when the frequency of operation is changed from the center frequency (13.1 GHz), the generated pencil beam(s) may slightly scan with frequency (according to the direction of the plane wave underneath the radiating slots and the 2D dispersion relation of the unit cell). The frequency response of the proposed PEX array is described in more detail in Supplementary Note 2.

**PEX array experiment.** A 7 × 7 PEX array prototype is fabricated using standard printed-circuit-board fabrication techniques. Notably, the size of the fabricated PEX array is 1/4 that of the full-wave simulated design. As a result, the measured gain values are expected to be at least 6 dB lower than the simulation results previously shown in Fig. 3. This smaller size is chosen to reduce the complexity and fabrication costs of both the PEX array and the required feeding network, for the proof-of-concept experimental demonstration presented here. A photo of the fabricated prototype is shown in Fig. 5a. The PEX array is experimentally characterized for both single and multiple beam generation modes, using a near-field scanning antenna characterization system as shown in Fig. 5b (details about the fabricated PEX antenna, the feeding network and the experimental procedure implemented are discussed in the "Methods" section). For single-beam generation, a single 1 × 16 feeding network is used to excite side A of the PEX array with a plane wave oriented along different angles ($\psi$) with respect to the x-axis. This generates pencil beams with different beam-pointing directions as illustrated in Fig. 6a, b, comparing the measured radiation patterns to those generated by a corresponding full-wave simulated 7 × 7 PEX array. Notably, the near-field scanning system uses an Az-over-El (Az/EL)

coordinate system, which is different from the Theta-Phi ($\theta$-$\phi$) system used in the full-wave simulation results[61]. It is possible to apply a series of coordinate system transformations and rotations to the measured results, and extract the $E$-plane and $H$-plane radiation patterns from them, however, this would involve many mathematical calculations, inevitably adding numerical errors to the measured results. Thus, it was preferred to plot the measured results directly using their native (Az/El) coordinate system, given that both ($\theta$-$\phi$) and (Az/El) spherical coordinate systems are identical along the cardinal planes ($x$-$z$ and $y$-$z$ planes). In particular, for the $x$-$z$ plane ($\phi = 0°$) the elevation angle ($\theta$) is

identical to Az, whereas for the $y$-$z$ plane ($\phi = 90°$) the elevation angle ($\theta$) is identical to El. Thus, both coordinate systems can be used interchangeably along or around the cardinal planes. Since, the proposed PEX antenna mostly scans the generated pencil beams close to these cardinal planes, the measured and simulated results can be compared directly even though they are technically plotted using different coordinate systems.

The measured far-field radiation patterns, corresponding to the case when side A is excited, are presented in Fig. 6a, b. Note that the Az angles, in this case, are measured along planes parallel to the $x$-$z$ plane with zero towards the $z$-axis, whereas the El angles are measured along the orthogonal planes that go through the $y$-axis with zero towards the $z$-axis (see Supplementary Note 4 for more details). Again, the electric-field polarization of the radiated beams is along the propagation direction of the excited plane wave, i.e., approximate along the $x$-axis. In any case, the PEX array is clearly able to generate independently-scannable pencil beams at broadside and other tilted angles along the El plane ($y$-$z$ plane), when side A is excited. The corresponding maximum deviation in the measured peak gain of the scanned pencil beams is around 1.28 dB. On the other hand, the maximum tilted-angle scan range achieved in Fig. 6 is for the case $\psi = \pm 20°$ and is around 33.5° away from broadside. The generated pencil beam was not scanned beyond that to avoid the generation of grating lobes, which often emerge for pencil beams generated at tilted angles that are bigger than a critical tilted angle ($\theta_c$). For the PEX unit cell shown in Fig. 2a, the critical tilted angle $\theta_c$ can be calculated at 13.1 GHz from $\theta_c = \sin^{-1}(\lambda_o/d - 1) = 37°$[56]. The dotted black circle in Fig. 3b corresponds to the calculated critical angle for the PEX unit cell, i.e., the grating-lobe limit. All the

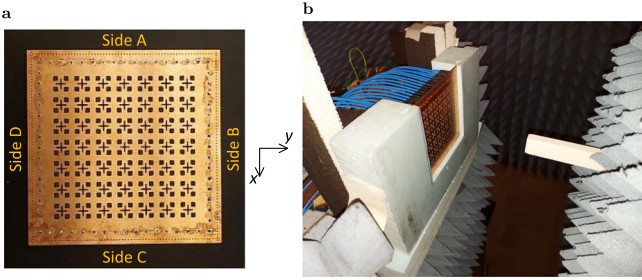

**Fig. 5 PEX array measurement. a** Top view of the fabricated 7 × 7 PEX antenna array prototype which is etched on a Rogers RT/duroid 5880 1oz substrate, showing the different sides of the array A, B, C and D, and the coordinate system used. (Dimensions = 131.4 mm × 131.4 mm × 1.575 mm). **b** A photo of the near-field scanning experimental setup used, showing the PEX prototype and the waveguide probe using to measure the near-field radiation from the antenna.

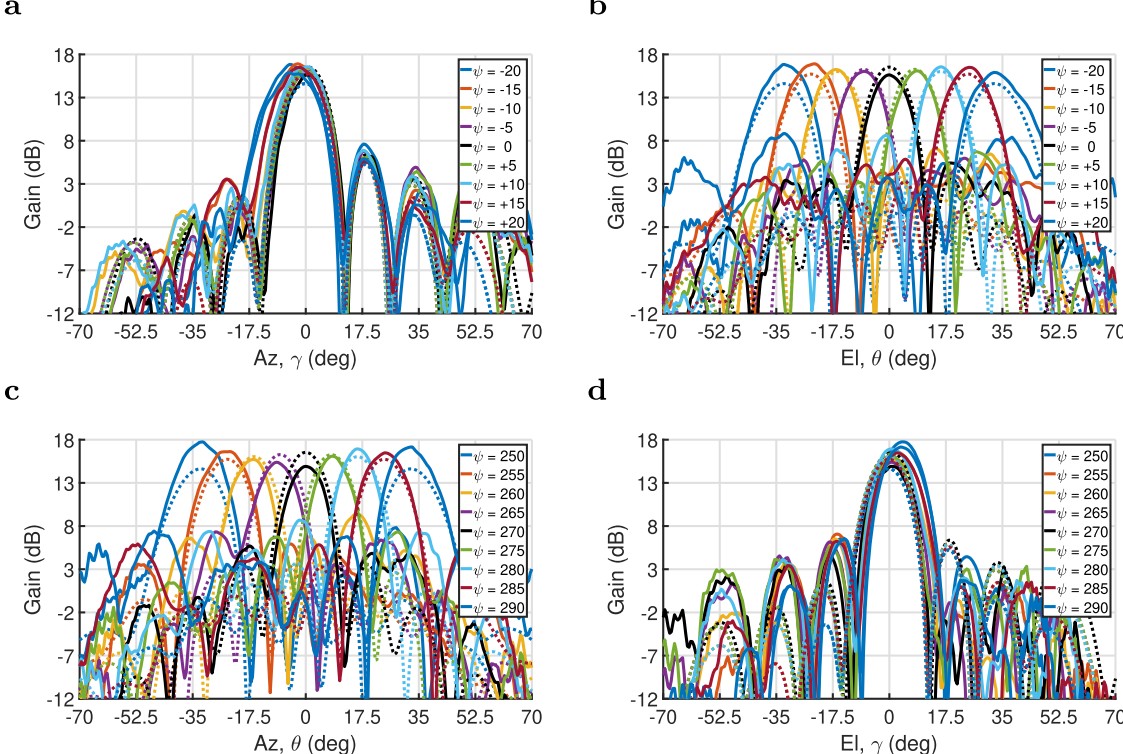

**Fig. 6 Single-beam measurement. a**, **b** Comparison between the measured (solid) and full-wave simulated (dotted) realized gain patterns (dB) at 13.1 GHz, along the **a** Az and **b** El directions for the measurement, and along the **a** $E$-plane and **b** $H$-plane for the simulation, for various values of ($\psi$), when only side A is excited. Successful electronic scanning is observed for the generated pencil beams in the El plane. **c**, **d** Comparison between the measured (solid) and full-wave simulated (dotted) realized gain patterns (dB) at 13.1 GHz, along the **c** Az and **d** El directions for the measurement, and along the **c** $H$-plane and **d** $E$-plane for the simulation, for various values of ($\psi$), when only side B is excited. Successful electronic scanning is observed for the generated pencil beams in the Az plane.

beam-pointing directions presiding inside this circle can be achieved without the generation of grating lobes as they satisfy the unimodal condition. Thus, this critical angle constitutes a practical limit on the maximum scan range possible from the proposed PEX antenna. Needless to say that these potential grating lobes can be entirely prevented by designing a PEX unit cell with a smaller physical size $d$, which in turn leads to a larger value for the critical tilted angle ($\theta_c$), and a larger dotted circle in Fig. 3b. This would allow more beam-pointing directions to satisfy the unimodal condition, and the generation of pencil beams along more of the possible beam-pointing directions predicted in Fig. 3b, without the generation of any grating lobes whatsoever.

On the other hand, side B of the PEX array is excited with a similar $1 \times 16$ feeding network, that excites plane-waves at various orientation angles with respect to the $y$-axis, and the polarization of the generated pencil beams is mostly orthogonal to the previous case of only exciting side A, i.e., is almost along the $y$-axis. Again, the PEX array is capable of generating a pencil beam along broadside and other tilted angles, this time along the Az plane ($x$-$z$ plane), as shown in Fig. 6c, d. The corresponding maximum deviation in the measured peak gain for the scanned pencil beams is around 2.84 dB. Although the proposed PEX antenna is symmetric, the beam scanning loss when side B is excited is significantly more than that of side A. This discrepancy can be attributed to imperfections in the measurement setup such as small unwanted bounces and reflections from the surfaces of the walls and tables surrounding the antenna, which are more severe farther away from broadside and may slightly differ for sides A and B.

Overall, it is observed that the measured beams are in good agreement with the full-wave simulation results, and they exhibit similar pencil-beam radiation angles as calculated from the analytical expressions and predicted by the full-wave simulations. Nevertheless, there is a discrepancy between the measured and simulated peak gain values in Fig. 6, even though the results correspond to a PEX antenna with the same physical size. This discrepancy can be explained by observing the phase-shifter' transmission response in Supplementary Note 1, where it is clear that the transmission amplitude changes as a function of the achieved phase shift. Hence, unlike the full-wave simulations, the fabricated PEX antenna is excited with progressively phase-shifted excitations with varying amplitudes. This causes a discrepancy between the measured and simulated peak gain values in Fig. 6. The maximum discrepancy between the measured and simulated peak gain when side A is excited is around 2.22 dB, and when side B is excited is around 3.13 dB. Needless to say that a better optimized phase shifter design, with a more constant transmission amplitude for different phase shifts, would allow a closer agreement between the measured and simulated peak gain values. In any case, the measured peak directivity achieved by the PEX antenna prototype is between 19.8–22.1 dB for the different measured cases, which is in good agreement with the full-wave simulation results, and further validates the proposed PEX design (see Supplementary Note 2 for more details).

Furthermore, the measured pencil beams exhibit an average of 2.1% 1 dB-bandwidth and 7.8% 3 dB-bandwidth when side A is excited, and an average of 3.3% 1 dB-bandwidth and 8.2% 3 dB bandwidth when side B is excited. It is observed that the measured bandwidths are slightly wider than the simulated values reported earlier in Table 1, even though the phase shifters used in the experiment have a frequency-dependent phase shift (see Supplementary Note 1), whereas the full-wave simulated PEX array is excited with a frequency invariant progressive phase shift. This can be partially justified by the fact that the measured PEX array is 1/4 of the size of the full-wave simulated version, hence, the measured pencil beams exhibit a much wider beamwidth, which can lead to a slightly wider directivity bandwidths. On the other hand, although the fabricated PEX antenna is symmetric, there is a very small discrepancy between the measured directivity bandwidth for the generated pencil beams from sides A and B. This is caused by the different dispersion characteristics of the phase shifters used to excite sides A and B, as well as small unwanted bounces and reflections around the measurement setup which may differ for the two sides.

From this discussion, it is observed that the proposed PEX array is clearly capable of scanning the generated pencil beams along both the azimuth and elevation planes, independently. These measured patterns demonstrate the versatility and flexibility of the proposed design, which can be considered as multiple antennas sharing the same radiating aperture, saving valuable real estate.

**Multiple-beam experiment**. It is also possible to generate multiple pencil beams by applying superposition to the PEX array, and exciting more than one plane wave simultaneously. For example, side A is excited with a plane wave that generates a broadside pencil beam, whereas side B is excited with a plane wave that generates a tilted pencil beam. Two $1 \times 16$ feeding networks and a single commercial $1 \times 2$ power splitter are used in this case. The measured radiation patterns for a descriptive case is depicted in Fig. 7a, e, where two pencil beams are successfully generated with the same beam-pointing directions as predicted earlier. It is also possible to generate two tilted pencil beams simultaneously as illustrated for the descriptive cases depicted in Fig. 7b–d, f–h. More measured cases of generating two pencil beams are shown in Supplementary Note 3. It is important to emphasize that there are no coupling or degradation effects between the generated pencil beams by sides A and B, as the individual beams exhibit orthogonal polarizations as discussed earlier, and the overall mutual coupling between the two orthogonal sides is significantly low for all the generated beams.

Obviously, superposition of even more plane waves is possible, and three or more pencil beams can be generated in principle. However, if it is required to generate multiple plane waves from the same side, the feeding network will be required to modulate the amplitudes of the individual port excitations, and not solely the phase, unlike the examples presented in this paper. All the results presented in the main body of the paper pertain to the center frequency of operation (13.1 GHz). When the frequency of operation is changed, the generated pencil beam(s) are expected to slightly scan as a result of the dispersion of the phase-shifters and the unit cell. This is discussed in more detail in Supplementary Note 2. The supplementary information also includes more details about the feeding networks in Supplementary Note 1.

Notably, the proposed antenna does not suffer from any current cancellation or mutual coupling problems from the closely-spaced peripheral sources. It is also clear that the proposed PEX antenna array exhibits numerous advantages such as the effective integration of multiple antennas into a single shared radiating aperture, which is able to generate single or multiple pencil beams at broadside and tilted radiation directions. For the multi-beam case, it is possible to operate all the individual beams generated from the different sides of the PEX array under transmit, or receive mode, or even under simultaneous transmit and receive, which leverages on the low mutual coupling between the different sides of the PEX array as shown in Supplementary Note 2. This has far-reaching potential applications, as the proposed design can be deployed in multiple-beam (MIMO) and

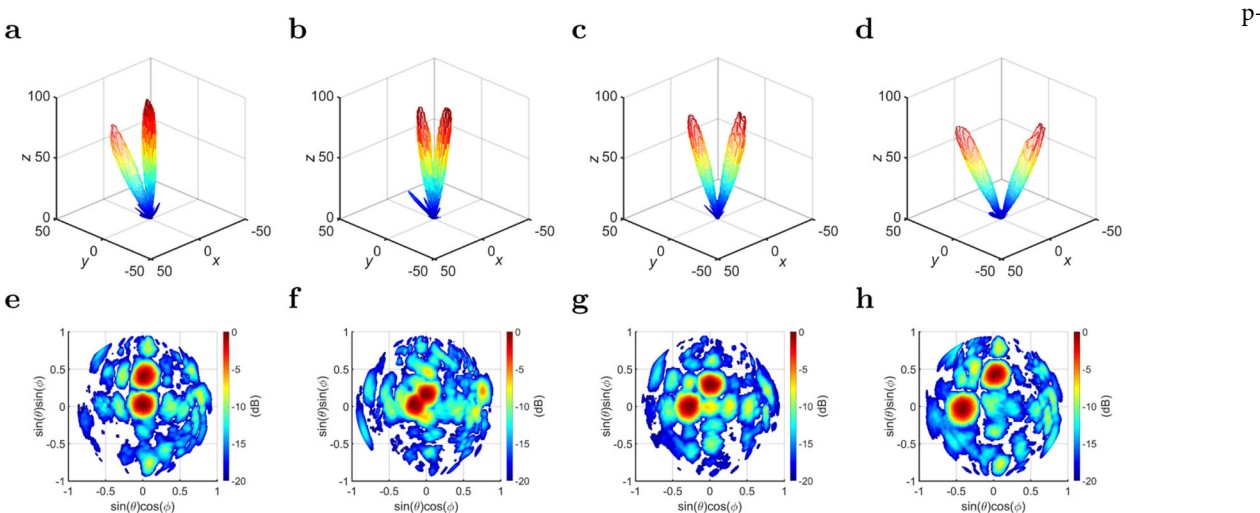

**Fig. 7 Multiple-beam measurement.** The normalized measured radiation patterns for various values of ($\psi$), when both sides A and B of the 7 × 7 PEX array are excited simultaneously: **a–d** The measured far-field 3D gain patterns for the cases: **a** $\psi = 0°$ and 285°, **b** $\psi = -5°$ and 275°, **c** $\psi = -10°$ and 280°, and **d** $\psi = -15°$ and 285°, plotted in linear scale (V/m) and normalized to an arbitrary value of 100. **e–h** The measured far-field U-V gain patterns for the cases: **e** $\psi = 0°$ and 285°, **f** $\psi = -5°$ and 275°, **g** $\psi = -10°$ and 280°, and **h** $\psi = -15°$ and 285°, plotted in dB scale. All the beams can be independently scanned, and a multitude of radiation directions are achieved. More measured results are presented in Supplementary Note 3.

duplex applications with simultaneous transmit and receive operation from the same antenna (not necessarily from the same direction). This ultimately comes at the price of the limited scan range of the generated pencil beams, which is capable of scanning only along predefined contours.

## Discussion
The proposed PEX array can continuously steer the generated pencil beam along a multitude of azimuthal and elevation directions. The PEX antenna is excited by plane waves which are synthesized directly by individual Huygens' sources with an electronically-controlled progressive phase shift that can be continuously tuned, and the sources are situated along the periphery of the radiating aperture according to the Huygens' equivalence principle. One of the advantages of the proposed PEX array is that it achieves a sizable reduction in the active-element count relative to traditional phased arrays, particularly for larger PEX arrays[45,46], as the active-element count now scales with the perimeter of the PEX array radiating aperture, instead of its area. There are numerous additional advantages of the proposed design, namely, the PEX array is capable of: (a) steering the plane-waves electronically without requiring any relative mechanical rotation between its plates or lenses (b) continuous spatial scanning as opposed to switched beams (c) exciting plane waves from all its sides leading to more scanned planes (when compared to alternative related antennas[41,43]) (d) generating a single or multiple independently-steered pencil beams. It is also possible to leverage the peripheral sources, by specifying suitable weights, to electronically tune the side-lobe levels of the generated pencil beams, and achieve some form of beamforming. This possibility will be investigated in the future. The proposed design can also be leveraged to generate multiple pencil beams from the PEX array simultaneously, thus mimicking the role of multiple independent antennas sharing the same radiating aperture, which saves valuable real-estate.

It is possible to extend the scan range further in the future by incorporating additional scanning techniques, such as placing the entire PEX array on a simple rotating pedestal that can be mechanically rotated. This will enable the full coverage of the

per hemisphere, where the elevation of the pencil beam is controlled by the PEX antenna, and a corresponding azimuth is set by the PEX antenna and fine tuned by the mechanical motor. Additionally, there are alternative techniques that can be envisioned to extend the scan range such as leveraging technologies that can change the effective permittivity of the medium filling the PEX cavity. Notably, the entire permittivity of the medium filling the PEX cavity is required to be changed in unison, i.e., the individual unit cells are not required to be individually tuned. One possibility to achieve this is by replacing the dielectric substrate with voltage-controlled materials such as ferroelectric materials[62], and controlling the permittivity of the material by DC biasing the ground and top plates of the PEX cavity. Another possibility is to fill the PEX cavity with an artificial dielectric such as a bed of nails going through holes in the ground plane, and mechanically controlling the effective permittivity by changing the height of nails going through the holes. These techniques and others can be implemented to extend the scan range of the PEX antenna to achieve full 2D spatial coverage.

For some applications, it is desirable to scale the proposed PEX antenna to higher millimeter-wave frequencies. Some of the challenges for that include additional material losses and realizing higher-frequency peripheral feeds. The proposed PEX array can be considered a substrate-integrated parallel-plate waveguide, thus, it is expected to exhibit similar material losses to typical substrate-integrated waveguide antennas at millimeter-wave frequencies. In addition, the losses at millimeter-wave frequencies are expected to be comparable to, or even less than, fully-fledged phased arrays that require inevitably more complicated feeding networks. It is also possible to envision fully-metallic versions of the proposed PEX array using an artificial dielectric such as a bed of nails to fill the PEX cavity[44], which can reduce the material losses significantly. On the other hand, the proposed PEX antenna, in its current form, exhibits coaxial connectors that are used to directly feed the radiating aperture at its periphery. At shorter wavelengths, these coaxial connectors might be too large to be closely placed together for proper PEX operation, nevertheless, this is not a fundamental limitation. Many substitutes for directly feeding the PEX array can be envisioned such as: (a) engineering transitions from spaced-out coaxial connectors to

narrowly-spaced microstrip/coplanar lines that feed the PEX array (b) integrating sources (transceivers) in IC form and placing them directly at the location of the peripheral feed points. Thus, from this discussion, it is clear that the proposed PEX antenna design can be potentially scaled to higher millimeter-wave frequencies with relative ease. On the other hand, expanding the frequency of operation to sub-THz range (100 GHz and beyond) could present challenges, when using the current PEX array implementation with phase shifters. Thus, successful PEX array operation can only be envisioned up to the millimeter-wave regime, if the phase shifters are not replaced with other higher-frequency suitable alternatives.

To conclude, in this paper, we propose a practical realization for the Peripherally-Excited (PEX) antenna concept. To realize the design, we specially engineer a radiating unit cell with a closed bandgap in the dispersion relation, and use it as the top per-forations for the PEX cavity. This enables the successful genera-tion of pencil beams at broadside and titled radiation directions from the PEX cavity. We show that the proposed structure is able to generate highly directive scannable pencil beams with high aperture efficiency. The proposed structure can also be operated in multiple modes, where single and/or multiple independent pencil beams are generated by simply controlling the excitation of the peripheral sources. Additionally, we show that the PEX antenna array is able to scan the generated beams along pre-defined contours with a versatile range of tilted angles. We experimentally validated the proposed structure using near-field antenna measurements techniques, and the experimental results are in good agreement with the full-wave simulations. This demonstrates the versatility of the proposed PEX antenna array, and confirms that the proposed design is a practical imple-mentation for the PEX antenna concept. The proposed design is quite simple to fabricate using standard printed-circuit-board fabrication techniques, and can be potentially scaled to higher millimeter-wave frequencies. This has numerous potential applications, and the design could perhaps be deployed in mulitple-beam (MIMO) and duplex applications with simulta-neous transmit and receive operation from the same antenna (not necessarily from the same direction). The proposed PEX array implementation is just the beginning of an alternative phased array technology with many future possibilities. It has many advantages such as the ability to generate single and/or multiple independently scanned pencil beams using only peripheral sources. Importantly, it achieves a sizable reduction in the active-element count especially for larger PEX arrays, compared with traditional phased arrays. Possible future implementations can be even potentially realized with an extended scan range, and operated at higher millimeter-wave frequencies which is highly desirable. Hence, it can potentially be deployed in some of the emerging systems that require the generation of scannable pencil beams, such as automotive radars, among others.

## Methods

**Radiating unit cell with closed bandgap**. The proposed unit cell is designed on a commercial Rogers RT/duroid 5880 1oz substrate with thickness = 1.575 mm. The unit cell is full-wave simulated at frequencies around the center frequency (13.1 GHz), using the "eigenmode" solver in Ansys HFSS[63]. The goal of the full-wave simulation is to optimize the dimensions of the slots, in order to close the bandgap at the Γ-point and enable successful radiation at and around broadside. To emulate an infinitely periodic array, master-slave periodic boundary conditions are applied along the sides of the unit cell, and a perfect matching layer (PML) is used at the top to absorb the generated radiation. In the design process, the dimensions of the cross-shaped slots are first determined to achieve a strong radiation from the corresponding leaky-wave mode, by realizing a large leakage constant and a low $Q$-factor. After that, the dimensions of the four square-shaped slots $S_1$ and $S_2$ are both optimized to achieve accidental degeneracy of the four eignemodes at the Γ-point (see Fig. 2a). In particular, the dimensions of the notches placed in the corner of the square-shaped slots are tuned, as they provide an additional degree of freedom, and

allow the independent control of the individual eigenmodes. After performing a parametric sweep analysis, the optimum slot dimensions are determined, and the optimized square-shaped slots are able to achieve the aforementioned accidental degeneracy of the eigenmodes and successfully close the bandgap at the Γ-point, as shown in the 2D dispersion relation (see Fig. 2b, c).

**PEX array simulation**. The full-wave simulation results of the $15 \times 15$ PEX array are performed using the "Driven Modal" solver in Ansys HFSS[63]. To realize the metallic side walls of the PEX cavity, an array of 0.8 mm metallic vias are placed, with a center-to-center separation of 2 mm. The PEX array is perforated at the top with a periodic array of the closed-bandgap unit cell (see Fig. 3a), and is capable of radiating at broadside and other tilted radiation directions. The peripheral Huy-gens' sources are implemented using PEC-backed coaxial feeding ports, which effectively behave as magnetic surface currents (see Fig. 1d). The distance between the coaxial feeds and the metallic side walls is optimized to minimize any reflec-tions from the peripheral ports and limit the mutual coupling between closely-spaced adjacent ports, as discussed in the paper (see Supplementary Note 2 for more information). The PEX array exhibits 31 coaxial ports along each of the four sides of the array, which are simulated using wave ports. There are also four dummy ports that are added in the four corners of the PEX array, used to eliminate any problems due to edge effects. These dummy ports are simply terminated to matched 50 Ω loads. The design parameters of the PEX array are $p = 7.15$ mm, $d = 14.3$ mm, $\epsilon_r = 2.2$, $d_p = 5.5$ mm, $t_p = 1.4$ mm and $h = 1.575$ mm. The effects of metallic and dielectric losses are included in these full-wave simulations, assuming a Rogers RT/duroid 5880 1oz substrate. For different simulation scenarios, some or all of the sides of the PEX array are excited with an appropriately phase-shifted excitation. This leads to the excitation of plane waves at various directions inside the PEX cavity, which in turn generates pencil beams at broadside and tilted radiation directions. Throughout the paper, all unexcited sides are effectively ter-minated to 50 Ω matched loads by the full-wave simulation. Overall, the proposed PEX array design is quite simple to simulate, is compatible with standard PCB fabrication techniques, and can be easily scaled to higher millimeter-wave frequencies.

**Phase-shifting feeding network**. There are various ways to implement the required phase-shifting feeding network. For instance, it is possible to integrate a feeding network with phase shifters to the back of the bottom side of the PEX array board, on additional layers that are bonded to its ground plane. In this case, metallic vias can be used to connected the phase shifters to the peripheral ports of the PEX array. This approach has the benefit of producing a low-profile single board that includes both the feeding network and the PEX array, with minimal external wiring required. However, it is not flexible enough to allow for experi-menting with different feeding networks or operating the PEX array under different operational modes, as the feeding network would be permanently bonded to the back of the PEX array. In this paper, it is preferred to follow a more flexible and modular approach where the feeding network(s) are designed and fabricated on separate board(s) from the PEX array. SMA-to-SMP RF coaxial cables are used to connect the feeding network board(s) to the different sides of the PEX array, depending on the mode of operation. This approach is more convenient for the proof-of-concept experimental demonstration in this paper. To properly excite the PEX antenna array, the feeding network includes a phase shifter for each peripheral source (port), hence, a $1 \times 16$ phase-shifting feeding network is designed for this purpose (see Supplementary Note 1 for more details). Two sample boards are fabricated, where each of them is sufficiently capable of entirely feeding one side of the PEX array prototype which contains only 15 peripheral ports (the extra port on the feeding network is unused). Each feeding network is used to excite one side of the PEX array with an electronically-controlled progressive phase-shifted excita-tion, which excites a plane wave underneath the radiating slots along various directions. The design and performance of the developed $1 \times 16$ phase-shifting feeding network is described in more detail in Supplementary Note 1. For the multiple-beam experiment, an additional $1 \times 2$ commercial power splitter is used in conjunction with two $1 \times 16$ feeding network boards to allow the excitation of sides A and B of the PEX array simultaneously.

**PEX array experiment**. The $7 \times 7$ PEX antenna array prototype is designed on a double-sided Rogers RT/duroid 5880 1oz substrate with thickness = 1.575 mm, and was fabricated by Candor Industries Inc and assembled by V.U.nics Inc. The fabricated PEX array prototype exhibits 15 peripheral feeding ports along each of the four sides of the PEX array, and four dummy corner ports that are added to eliminate any problems with edge effects. Depending on the mode of operation (single or multiple-beam generation), one or more of these sides are excited with individually phase-shifted excitations, whereas the dummy corner ports are always terminated to matched loads. SMP coaxial connectors are used to excite the individual peripheral ports since they exhibit a small physical size, thus, they can be placed sufficiently close together, compared to other options such as standard SMA coaxial connectors which are slightly bigger in size.

The fabricated $7 \times 7$ PEX array prototype is experimentally characterized using a planar near-field scanning NearField Systems Inc. (NSI) antenna measurement system, which uses an Agilent Technologies N5244A vector network analyzer

(VNA) (see Fig. 5b). The NSI system is able to convert the measured near-field data to far-field radiation patterns. In the experimental results presented in the paper, some sides of the PEX array are excited with an appropriately phase-shifted excitation, whereas the remaining unexcited sides are terminated to 50 Ω matched loads to absorb any remaining unradiated power. The PEX array is experimentally characterized for both single and multiple-beam generation modes. It is worth noting that the NSI measurement system uses an Az-over-El (Az/EL) coordinate system which is different from the Theta-Phi ($\theta$-$\phi$) system used in the full-wave simulation results[61] (see Supplementary Note 4 for more details). Thus, it is very challenging to plot the E-planes and H-planes radiation patterns from the measured results similar to the full-wave simulations. Instead, the Az and El measured radiation patterns are presented in the paper (see Fig. 6a–d). An A-Info LB-10180 1–18 GHz wide-band horn antenna is used to calibrate the NSI measurement system, by implementing a standard gain comparison procedure, which allows the experimental characterization of the far-field realized gain patterns generated by the fabricated PEX antenna prototype. Additionally, the losses and reflections due to the feeding network and RF cables are calibrated out from the measured radiation patterns, by characterizing their performance separately, and removing their effect from the measured patterns using simple post processing techniques.

## Data availability

All key data generated and analyzed are included in this paper and its supplementary information. Additional data sets that support the plots within this paper and other findings of this study are available from the corresponding author upon reasonable request.

## Code availability

The codes and simulation files that support the plots and data analysis within this paper are available from the corresponding author upon reasonable request.

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

## Acknowledgements

This work was sponsored by the Natural Sciences and Engineering Research Council of Canada (NSERC) and a Canada Research Chair (Tier 1). The authors would like to acknowledge the help provided by Professor Sean V. Hum and Nicolas Faria for providing access to their near-field antenna testing facility. The authors would also like to acknowledge the extremely useful discussions with Michael Chen, Amirmasoud Ohadi and Minseok Kim.

## Author contributions

A.H.D. performed formulation, analysis, simulations, physical design, experiments and generation of the results, and G.V.E. supervised all these stages. A.H.D. and G.V.E. contributed to conceiving the idea, and writing and editing the manuscript.

## Competing interests

The authors declare no competing interests.
