## [Peer Review File · Nature Communications]

REVIEWER COMMENTS

Reviewer #1 (Remarks to the Author):

This is a high quality work that provides the first experimental demonstration of a recently proposed concept, namely Peripherally-Excited Antenna Arrays. The designs developed are novel and importance for the engineering as well as physics/optics community. The results validate the PEX array concept and open up new directions of R&D for future antennas in 5G (and beyond) and satellite communication systems.

The technical information in the paper is correct, but it can be more complete if the following issues are addressed:

- 2D plots of dispersion diagrams can be also presented in Fig. 2 showing clearly the closed band gaps
- What is the maximum range of beam scanning that can be achieved? Is the max value for $\psi=20^\circ$? Can it be more and how?
- Bandwidth and beam squinting effect limitations of the proposed design should be explained clearly. A number of radiation pattern graphs are provided in the supplementary file, however it would be useful to discuss these limitation in the main text of the paper.
- What is the overall efficiency of the antenna design? Only aperture and radiation efficiency values are mentioned. It seems that 44% of the energy goes from one side to the opposite one. What does this mean for the overall efficiency of the proposed antenna?
- In the measured results, specify how many dBs exactly is the beam scanning loss , as this is not clear from the graphs.
- The measured gain values are much lower than the simulations. Please clarify this difference in numbers and explain the main reasons for this discrepancy this in more detail.

Reviewer #2 (Remarks to the Author):

Please find below my comments, section by section. The review report is then completed with some general remarks.

Introduction

The state-of-the-art is quite complete and it presents sparse or thinned arrays as possible solutions to reduce the number active elements in phase arrays. However, although reflector antennas are limited to "fixed beam generation," multiple sources in the focal plane do enable scanning by beam-switching (even if over a limited field of view). Besides, single-layer [A] or multi-layer [B] low-profile quasi-optical beam-formers can be used to generate the plane-wave in a parallel plate waveguide by activating different sources in the focal plane. The introduction could be modified to take these aspects into account.

[A] Y. J. Cheng, W. Hong, K. Wu and Y. Fan, "Millimeter-wave substrate integrated waveguide long slot leaky-wave antennas and two-dimensional multibeam applications," IEEE Trans. Antennas Propag., 59, 40-47 (2011).

[B] M. Ettorre, R. Sauleau and L. Le Coq, "Multi-beam multi-layer leaky-wave SIW pillbox antenna for millimeter-wave applications," IEEE Trans. Antennas Propagation, 59, 1093-1100 (2011).

In line 67, the authors have written "the proposed antenna (...) can be easily scaled to higher millimeter-wave frequencies." In my opinion, several hindrances may be encountered when up-scaling this design. For instance, the footprint of the SMP connectors might not be sufficiently small to enable the excitation of the individual peripheral ports at shorter wavelengths. Besides, the losses can be very high beyond Ka-band.

Could the authors comment on the highest frequency that one could achieve with the proposed PCB approach?

In line 72, the authors have written "deployed in ... duplex applications." Duplex operation is discussed later when the multi-beam results are presented. I have a question on this regard. In full-duplex operation mode, the antenna should point at the same direction when transmitting and receiving. Therefore, the beam-coupling factor (Stein, IRE Trans. Antennas Propag., 10(5), 548-557, 1962) will be high and the cross-coupling between sides might increase. Have the authors investigated this case?

2. Concept of the PEX antenna array

In Fig. 1a, the authors show "Love's equivalence principle" for PEC material, which is commonly known as Schelkunoff equivalence principle in the Electromagnetics community. This is just a matter of wording, not really important. Otherwise, the concept is clearly explained and well-documented in previous publications.

3. Results

In the paragraph *Radiating Unit Cell with Closed Bandgap*, the authors present the unit cell employed to achieve accidental degeneracy of the eigenmodes at the Gamma point. From the results, it is clear that the unit cell achieves the desired behavior. However, this unit cell is different from that in [23], [51] and [52]. Is the design solely based on the optimization process described in the Methods section or have the authors built on transmission line theory as in [23] or used another theoretical starting point?

In the paragraph *Analytical Beam-Pointing Directions*, the authors use ϵ_e in (1a) to (2b). However, ϵ_r was used for the formulae in [38]. Could the authors explain this difference? In my opinion, it would be useful to motivate the definition of ϵ_e in page 8.

Fig. 3b shows the achievable beam pointing direction. However, it is not clear to me the range of ψ angles used to obtain the lines for the different modes. If I have understood correctly, the empty regions in the visible region in Fig. 3b correspond to directions where pointing cannot be achieved. Does this mean that for electrically larger antennas (narrower beams) the portion of the visible region covered by the generated beams will be reduced?

In the paragraph *PEX array simulation*, the authors report a high simulated aperture efficiency of 84.6% for a $10\lambda \times 10\lambda$ aperture. This means that the obtained directivity is quite close to the ideal one of approximately 31 dB. However, the realized gain is only 25.1 dB due to the relatively large (44%) amount of power delivered from Side A to Side C. In other words, the almost uniform aperture distribution is obtained at the expense of coupling power to the opposite side. The authors explain that "a bigger PEX array will exhibit less power lost to the resistive terminations along side C." In my opinion, they should also explain that with the uniform (as opposed to amplitude-tapered) radiating structure, the reduction of power lost to resistive terminations in larger PEX arrays will lead to a reduction of the aperture efficiency. I believe that this limitation should be mentioned in the introduction too, since it has an impact on the practical application of this concept.

In Table 1, could the authors report for each case the excited side and the power coupled to the other sides? In my opinion, this information can be of interest for the reader.

Fig. 4 shows 3D radiation patterns for different multi-beam configurations. I believe that color maps in the UV plane would be clearer. If the authors choose to leave the figure as is, please include at least the color bar to read the values. Also, which magnitude are the authors plotting? Please explain. The same comments apply to Fig. 7.

The paragraph *PEX Array Experiment* reports measurements carried out by a planar near-field scanner from NSI. I am not an expert on antenna measurement. However, I do not understand why it is challenging to plot the E- and H-plane. Using the fields sampled on the plane by the near-field probe, one can apply a near- to far-field transformation to obtain the far-fields at the desired θ - ϕ pairs. Could the authors explain me with more detail? I am just curious, as long as the compared planes are the same, the

authors can keep the preferred reference system.

In line 248, please replace "patters" with "patterns."

In Fig. 6, could the authors comment on the difference between simulated and measured patterns? In Figs. 6b-c, the measured patterns for high elevation angles are more directive than the ones predicted by full-wave simulations, whereas the measured directivities for zero elevation are lower than the predicted ones. Also, in line 258, I think the figure shows a "good agreement" rather than a "great agreement."

In line 282, is there any particular reason for choosing 13 GHz as center frequency? Do the authors have a particular application in mind?

Also in line 282, I think that further detail should be included in this paragraph about the bandwidth of the proposed array. In the supplementary material (supplementary Figs. 6-7), the authors show a quite stable beam between 12.5 GHz and 13.7 GHz, is this the achievable bandwidth? I suggest to choose a criteria (1 or 3 dB directivity drop) to give some tentative numbers for the relative bandwidth.

References

In the references section, [46] duplicates [38]

Concluding remarks

The authors present the experimental validation for the Peripherally Excited (PEX) array concept. This concept has been recently presented by the authors in some of their previous works [38-43]. Some preliminary steps towards the experimental demonstration had been already taken in [41] and [43] without considering the radiating part of the system. Here, the authors build on their previous works [51]-[52] to design this radiating part and the Huygens sources modifying the structure in [38], Fig. 10. I agree with the authors that the reported experiments have not been shown before, but I suggest to present with more clarity the evolution with respect to previous developments. In other words, the novel contributions and their impact should be highlighted.

On the other hand, although the authors report the limitations of the system in terms of the achievable pointing directions, other important limitations are somehow overlooked. For instance, the trade-off between power coupled to the opposite sides and the aperture efficiency should be discussed. Likewise, some guidelines about the achievable bandwidth should be introduced in the paper. These are important aspects that can hinder the application of the proposed antenna system for MIMO and automotive radar applications and the impact of the proposed concept on the field.

Otherwise, I think that the paper is complete for understanding and technically correct. Except for the points mentioned above, the theoretical and experimental results of the paper accompanied by sufficient analyses, discussions and justifications.

Reviewer #3 (Remarks to the Author):

The paper presents the practical implementation and experimental validation of peripherally excited array antenna, a concept previously introduced by the same authors. It basically consists in an array of slots excited by a slow wave propagating in the parallel plate waveguide formed by the slot plate and the ground plane. The slow wave is excited by sources located at the periphery of the radiating aperture, and its linear phase front is rotated by properly selecting the phase of the sources. Radiation is achieved through the interaction of the slow wave with the periodic perturbation introduced by the slots, according to a leaky wave mechanism, for which the radiating beam direction is determined by the relative orientation of the slow wave wavefront and the periodicity axes. Consequently, the radiated beam is scanned along a certain 1D path in the uv plane when the orientation of the wave front of the exciting wave is changed.

The paper is clear and well written, the proposed development is interesting, and it contains sufficient novelty to deserve publication. However, some aspects of the proposed solution should be better clarified, and references to similar solutions should be added.

In particular, the scanning mechanism of the proposed solution is the same of the so-called "series fed continuous transverse stub", or CTS, in which the rotation of the phase front is usually achieved through mechanical rotation. More recent works have proposed the use of switching sources exciting a rotationally symmetric planar lens. A few relevant references are suggested below

- W. Milroy, "The continuous transverse stub (CTS) array: basic theory, experiment, and application," in 1991 Antenna Applications Symposium, 1991.
- <https://www.thinkom.com/technology/>
- K. Tekkouk, J. Hirokawa, R. Sauleau, and M. Ando, "Wideband and large coverage continuous beam steering antenna in the 60-GHz band," *IEEE Trans. Antennas Propag.*, vol. 65, no. 9, pp. 4418–4426, 2017.
- Y. B. Li, R. Y. Wu, W. Wu, C. B. Shi, Q. Cheng, and T. J. Cui, "Dual physics manipulation of electromagnetic waves by system-level design of metasurfaces to reach extreme control of radiation beams," *Adv. Mater. Technol.*, vol. 2, no. 1, p. 1600196, 2017.
- J. Ruiz-García, E. Martini, C. D. Giovampaola, D. González-Ovejero and S. Maci, "Reflecting Luneburg Lenses," early access in *IEEE Transactions on Antennas and Propagation*.

Using this mechanism, full coverage of the upper hemisphere can be only achieved by adding a mechanical rotation of the radiating aperture. Although this is mentioned in the paper in the section entitled "analytical beam pointing direction", such a limitation should be more clearly explained already in the Introduction. In fact, this is the price to be paid for the complexity reduction in the feeding network: controlling only the phases of the elements at the periphery of the aperture instead of the phases throughout the aperture implies that the beam can only be scanned along a curved path instead of covering a conical region. As a matter of fact, the restricted achievable coverage appears to be a severe limitation for the application of the proposed solution in MIMO. Also, the angles reported in eq. 2, and the presentation in Fig. 3b, which shows the angles relevant to all the possible Floquet modes, are misleading, since they fail to account for the unimodal condition. The practically usable modes for the claimed objective to obtain a high directivity pencil beam are only the ones that can be individually excited, as the ones identified by dots in Fig. 3b.

Concerning the presentation of the results, some information could be added to increase clarity (some of them are included in the Supplementary material, but it would be better to have them also in the main body of the paper):

- overall efficiency of simulated and measured antennas should be indicated (for simulations, this can actually be reconstructed by elaborating the information provided for the single contributions, but it would be nice to have it directly available), and it should be specified if it includes the contribution of the feeding network.
- the possible causes of the few dB discrepancies between the gain values for simulation and measurements should be discussed .
- it seems that the unit cell design is only based on the real part of the propagation constant. Is the leaky wave radiation constant also considered in the design?
- It is mentioned that the proposed antenna could be used for duplex applications with simultaneous transmit and receive operation from the same antenna. In this case, how would the coupling between sources on different sides affect the performances?
- Are the quantities compared in Fig. 6 consistent, in spite of the different reference systems? It is not completely clear why having the data available at all angles from near-field to far-field transformation it is not possible to get for the measurements the same field cut available from simulations.

Other minor comments:

- The following reference could be added to explain the scanning mechanism and the derivation of eqs. 1a and 1b, not so easy to understand from the paper.

A. Bhattacharyya, "Theory of Beam Scanning for Slot Array Antenna Excited by Slow Wave," in *IEEE Antennas and Propagation Magazine*, vol. 57, no. 2, pp. 96-103, April 2015.

- In the caption of Fig. 3b "crosses" should be replaced by "circles" or "dots"
- From the plots reported in the Additional material, it seems that beam scanning is more significant for

measurements than for simulations. However, this could be due the different reference systems.

- In section 4 is stated that "the PEX antenna array is able to scan the generated beams along predefined azimuthal planes". In this context, what is actually meant by "azimuthal"?

- Throughout the paper there are some repetitions, due to the replication of some pieces of information in the Results and Method sections.

Reviewer #1:

We have carefully considered every comment in the Reviewers' reports. We believe that these comments helped to improve our manuscript, and we would like to thank the reviewers for that. We would also like to thank the reviewers for the timely handling of our manuscript.

This is a high quality work that provides the first experimental demonstration of a recently proposed concept, namely Peripherally-Excited Antenna Arrays. The designs developed are novel and importance for the engineering as well as physics/optics community. The results validate the PEX array concept and open up new directions of R&D for future antennas in 5G (and beyond) and satellite communication systems.

We thank the reviewer for the positive disposition on our paper.

The technical information in the paper is correct, but it can be more complete if the following issues are addressed:

- 2D plots of dispersion diagrams can be also presented in Fig. 2 showing clearly the closed band gaps

A 2D contour plot of the dispersion relation around the Γ -point has been added to Fig. 2c clearly showing the closed bandgap. This figure is introduced in the revised paper in line **218**.

- What is the maximum range of beam scanning that can be achieved? Is the max value for $\psi=20^\circ$? Can it be more and how?

The maximum tilted-angle scan range achieved in Fig. 6 is indeed for the case $\psi = \pm 20^\circ$ and is around 33.5° away from broadside. The generated pencil beam was not scanned beyond that to avoid the generation of grating lobes, which often emerge for pencil beams generated at tilted angles that are bigger than a critical tilted angle (θ_c). For the PEX unit cell shown in Fig. 2a, the critical tilted angle θ_c can be calculated at 13.1GHz from $\theta_c = \sin^{-1}(\lambda_0/d - 1) = 37^\circ$. A dotted black circle has been added to Fig. 3b showing the calculated critical angle for the PEX unit cell, i.e. the grating-lobe limit. All the beam-pointing angles presiding inside this circle can be achieved without the generation of grating lobes as they satisfy the unimodal condition. Thus, this critical angle constitutes a practical limit on the maximum scan range possible from the proposed PEX antenna. Needless to say that these potential grating lobes can be entirely prevented by designing a PEX unit cell with a smaller physical size d , which in turn leads to a larger value for the critical tilted angle (θ_c), and a larger dotted circle in Fig. 3b. This would allow more beam-pointing directions to satisfy the unimodal condition, and the generation of pencil beams along more of the possible beam-pointing directions predicted in Fig. 3b, without the generation of any grating lobes whatsoever. This discussion has been added to the paper in lines **234** and **392**.

- Bandwidth and beam squinting effect limitations of the proposed design should be explained clearly. A number of radiation pattern graphs are provided in the supplementary file, however it would be useful to discuss these limitation in the main text of the paper.

Indeed, the proposed PEX antenna array achieves radiation by exciting a travelling leaky-wave mode, hence, the generated pencil beams undergo a slight spatial scanning as the frequency of operation is changed. The 1dB and 3dB bandwidths achieved by the generated pencil beams at the center frequency (13.1GHz) for different plane-wave directions (ψ) have been added to Table 1. For example, at broadside, the generated pencil beam achieves a 2.5% 1dB-bandwidth and a 4.5% 3dB-bandwidth. Note that this bandwidth limitation is caused by the spatial scanning of the generated pencil beams, and not beam degradation. Being a LWA, the proposed PEX array can in principle be operated over a wider bandwidth while maintaining a directive beam, without beam degradation (as shown in supplementary section S2). This discussion has been added to the paper in line **306**.

- What is the overall efficiency of the antenna design? Only aperture and radiation efficiency values are mentioned. It seems that 44% of the energy goes from one side to the opposite one. What does this mean for the overall efficiency of the proposed antenna?

The proposed PEX antenna array, at its heart, is a uniform leaky-wave antenna, thus, it experiences an inevitable tradeoff between aperture efficiency, radiation efficiency, and termination efficiency. The latter refers to any remaining unradiated power that is lost to the resistive terminations. In Table 1, a breakdown of the different portions of power that reach the 4 sides of the PEX array when side A is excited are shown, together with the corresponding aperture and radiation efficiencies. This power breakdown does not include the effects of the feeding network that will be later used to excite the PEX array in the experiment. For the broadside case, it is observed that 44.4% of the incident power at side A reaches the opposite side C. Note that a larger PEX array will exhibit more radiation from the leaky wave, and less power will reach side C as a result, however, the aperture efficiency will inevitably drop as well. To mitigate this tradeoff, it is possible to design a PEX antenna array with tapered radiation from the unit cells, which is a standard practice for leaky-wave antennas that aim at achieving high aperture and radiation efficiencies simultaneously. This discussion has been added to the paper in lines **278** and **298**.

- In the measured results, specify how many dBs exactly is the beam scanning loss, as this is not clear from the graphs.

The maximum deviation in the measured peak gain for the generated pencil beams when side A is excited is around 1.28dB, and when side B is excited is around 2.84dB. Although the proposed PEX antenna is symmetric, the beam scanning loss for the second case is significantly more than the first case, which can be attributed to imperfections in the measurement setup such as small unwanted bounces and reflections from the surfaces of the walls and tables surrounding the antenna, which are more severe farther away from broadside and may slightly differ for sides A and B. Comments have been added to the paper in lines **391** and **413**.

- The measured gain values are much lower than the simulations. Please clarify this difference in numbers and explain the main reasons for this discrepancy this in more detail.

The measured gain values in Fig. 6 are significantly lower than the simulation results shown in Fig. 3. This is simply because the size of the fabricated PEX array is 1/4 that of the simulated design. Hence, the measured gain values are expected to be at least 6dB lower than the simulations. A comment has been added in the paper in line **358**.

On the other hand, there is a discrepancy between the measured and simulated peak gain values in Fig. 6, even though the results correspond to a PEX antenna with the same physical size. This discrepancy can be explained by observing the phase-shifters' transmission response in supplementary section S1, where it is clear that the transmission amplitude changes as a function of the achieved phase shift. Hence, unlike the full-wave simulations, the fabricated PEX antenna is excited with a progressively phase-shifted excitation with various amplitudes. This causes a discrepancy between the measured and simulated results in Fig. 6. The maximum discrepancy between the measured and simulated peak gain when side A is excited is around 2.22dB and when side B is excited is around 3.13dB. Needless to say that a better optimized phase shifter design with a more constant transmission amplitude for different phase shifts would allow a closer agreement between the measured and simulated peak gain values. This discussion has been added to the paper in line **421**.

Reviewer #2:

We have carefully considered every comment in the Reviewers' reports. We believe that these comments helped to improve our manuscript, and we would like to thank the reviewers for that. We would also like to thank the reviewers for the timely handling of our manuscript.

Please find below my comments, section by section. The review report is then completed with some general remarks.

Introduction

The state-of-the-art is quite complete and it presents sparse or thinned arrays as possible solutions to reduce the number active elements in phase arrays. However, although reflector antennas are limited to "fixed beam generation," multiple sources in the focal plane do enable scanning by beam-switching (even if over a limited field of view). Besides, single-layer [A] or multi-layer [B] low-profile quasi-optical beam-formers can be used to generate the plane-wave in a parallel plate waveguide by activating different sources in the focal plane. The introduction could be modified to take these aspects into account.

[A] Y. J. Cheng, W. Hong, K. Wu and Y. Fan, "Millimeter-wave substrate integrated waveguide long slot leaky-wave antennas and two-dimensional multibeam applications," IEEE Trans. Antennas Propag., 59, 40-47 (2011).

[B] M. Ettorre, R. Sauleau and L. Le Coq, "Multi-beam multi-layer leaky-wave SIW pillbox antenna for millimeter-wave applications," IEEE Trans. Antennas Propagation, 59, 1093-1100 (2011).

Indeed, continuous transverse stub arrays and switched-beam antennas are relevant to our proposed design. The following has been added to the introduction starting from lines **48, 64, 88, 91, 100, 237, 289** and **489**.

An interesting alternative antenna concept to phased arrays is the "Continuous Transverse Stubs (CTS) Array" [C1]. This concept does not require phase shifters, instead, it typically relies on some relative mechanical rotation between its constituent components [C2]. The CTS array usually comprises a parallel-plate waveguide perforated at the top with a 1D array of long slots, which are excited with a plane wave orthogonal to the long side of the slots. The slots radiate a pencil beam in free space through a travelling leaky-wave antenna (LWA) operation. The generated pencil beam can be scanned in both elevation and azimuth along predefined paths, by mechanically changing the relative orientation between the incident plane wave and the 1D slots. In [C2], this CTS array concept has been realized using a fully-metallic structure with three separate metal layers. These metal layers are implemented using contactless gap waveguides, allowing the mechanical rotation of the top radiating layer relative to the bottom feeding layers, thus changing the relative orientation between the excited plane wave and the 1D slots. This leads to a continuous scanning of the generated pencil beam in both azimuth and elevation along a single predefined path. Note that the 1D slots can only be excited from a single side, which limits the possible scan planes from the structure. To achieve full-space scanning, it is required to mechanically rotate the three metal layers together, in addition to an independent relative rotation between them.

On the other hand, numerous switched-beam antenna designs have been proposed [A,B,D1,D2] that are operated in a similar manner to the CTS array, and are able to combat the aforementioned tradeoff between scannability and cost. These antenna designs also eliminate the need for phase shifters, and are able to change the relative orientation between the excited plane wave and the radiating structure by using single-layer [A,D1] or multi-layer [B,D2] switched multi-port beamforming networks (lenses). These networks exhibit multiple switched-input ports situated along the focal plane/region of a lens such as a parabolic or Luneburg lens, and they leverage the Fourier-transform properties of the lens to excite a LWA structure with switched plane waves, by activating the different elementary input sources. Thus, this LWA structure is operated using a switched-beam operation, and each input port corresponds to a discrete pencil beam in space situated along predefined paths. However, one difficulty with these switched-beam antennas is that they cannot scan the generated pencil beams continuously, which is the price of the switched-beam operation, and usually the 3-dB beamwidth of the generated beams needs to be carefully optimized for an overlapping spatial coverage. This becomes more challenging when very narrow beams are required. Another potential drawback is that some lens designs such as parabolic reflectors are inherently constraint to reduced spatial scanning (angular coverage), due to eventual aberrations and spillover loss from feed elements displaced farther from the focal point. Also, these switched-beam antennas are typically not designed for broadside operation, which is inconvenient in some applications. More importantly, the LWA unit cells in [A,B,D1,D2] are only excited from along a single side of the LWA, and are not excited from the orthogonal side (although it is possible in principle), which limits the number of possible scan planes from these switched-beam antennas. In any case, a LWA unit cell that is optimized for broadside operation, while allowing excitation from the two orthogonal sides of the LWA is highly desirable, and can in principle double the number of possible scan planes from the LWA, effectively constituting multiple antennas in one shared radiating aperture.

In a nutshell, the aforementioned CTS and switched-beam antennas either require mechanical rotation or input-port switching to change the relative orientation between the excited plane wave and the slot array. In contrast, for the proposed PEX antenna design, the plane waves are synthesized directly by individual Huygens' sources with an electronically-controlled progressive phase shift that can be continuously tuned, and the sources are situated along the periphery of the radiating aperture according to the Huygens' equivalence principle. Thus, the proposed PEX unit cell is capable of generating electronically-steered pencil beams at and around broadside. Also, the symmetric nature of the unit cell enables the excitation of the PEX array from all its sides, enabling the generation of plane waves at arbitrary orientations inside the PEX cavity (the plane wave can be excited from any direction along any angle), which leads to more possible scanned planes from the radiating aperture (at least double the scanned planes of typical CTS structures). This capability can also be leveraged to generate multiple pencil beams from the PEX array simultaneously.

It is important to highlight that the PEX array can scan the generated beam in azimuth and elevation similar to the aforementioned CTS and switched-beam antennas. However, there are numerous differences between both concepts, namely, the PEX array is capable of: (a) steering the plane-waves electronically without requiring any relative mechanical rotation between its plates or any cumbersome lenses (b) continuous spatial scanning as opposed to switched beams (c) exciting plane waves from all its sides leading to more scanned planes (d) generating a single or multiple independently-steered pencil beams. It is also worth highlighting that it is possible to leverage the peripheral sources, by specifying suitable weights, to electronically tune the side-lobe levels of the generated pencil beams, and achieve some form of beamforming. This possibility will be investigated in the future.

In line 67, the authors have written “the proposed antenna (...) can be easily scaled to higher millimeter-wave frequencies.” In my opinion, several hindrances may be encountered when up-scaling this design. For instance, the footprint of the SMP connectors might not be sufficiently small to enable the excitation of the individual peripheral ports at shorter wavelengths. Besides, the losses can be very high beyond Ka-band. Could the authors comment on the highest frequency that one could achieve with the proposed PCB approach?

Some of the challenges of operating the proposed PEX antenna at higher millimeter-wave frequencies include additional material losses and realizing higher-frequency peripheral feeds. The proposed PEX array can be considered a substrate-integrated parallel-plate waveguide, thus, it is expected to exhibit similar material losses to typical substrate-integrated waveguide antennas at millimeter-wave frequencies. In addition, the losses at millimeter-wave frequencies are expected to be comparable to, or even less than, fully-fledged phased arrays that require inevitably more complicated feeding networks. It is also possible to envision fully-metallic versions of the proposed PEX array using an artificial dielectric such as a bed of nails to fill the PEX cavity [D2], which can reduce the material losses significantly. On the other hand, the PEX antenna, in its current form, exhibits coaxial connectors that are used to directly feed the radiating aperture at its periphery. At shorter wavelengths, these coaxial connectors might be too large to be closely placed together for proper PEX operation, nevertheless, this is not a fundamental limitation. Many substitutes for directly feeding the PEX array can be envisioned such as: (a) engineering transitions from spaced-out coaxial connectors to narrowly-spaced microstrip/coplanar lines that feed the PEX array (b) integrating sources (transceivers) in IC form and placing them directly at the location of the peripheral feed points. This discussion has been added to the paper in line **523**.

In line 72, the authors have written “deployed in ... duplex applications.” Duplex operation is discussed later when the multi-beam results are presented. I have a question on this regard. In full-duplex operation mode, the antenna should point at the same direction when transmitting and receiving. Therefore, the beam-coupling factor (Stein, IRE Trans. Antennas Propag., 10(5), 548-557, 1962) will be high and the cross-coupling between sides might increase. Have the authors investigated this case?

In the paper, we mention the possibility of using the proposed PEX antenna for simultaneous transmit and receive (duplex) operation, but not necessarily from the same direction. Indeed, one of the main challenges in achieving multiple-beam or duplex operation from any antenna is the mutual coupling that may occur between the different beams, which limits the radiation efficiency of the antenna and prevents the successful duplex operation (Stein, IRE Trans. Antennas Propag., 10(5), 548-557, 1962). To minimize this unwanted mutual coupling, it is required that the transmitted and received beams exhibit a low beam-coupling factor, i.e. the beams are required to be orthogonal. The proposed PEX antenna achieves this by exhibiting orthogonal polarizations for the beams generated from sides A and B, as the polarization of the generated pencil beams is along the direction of propagation of the excited plane wave inside the PEX cavity (which is orthogonal for sides A and B). Thus, this causes a remarkably low mutual coupling between the orthogonal sides of the PEX array, and there are no coupling or degradation effects between the generated beams by sides A and B (see Table 1 and supplementary section S2), which enables the successful multiple-beam and duplex operation of the proposed PEX array. This discussion has been added to the paper in lines **116, 250, 290, 332, 388, 409, 462, 485** and **557**, and the paper now mentions duplex instead of fully duplex operation.

2. Concept of the PEX antenna array

In Fig. 1a, the authors show “Love’s equivalence principle” for PEC material, which is commonly known as Schelkunoff equivalence principle in the Electromagnetics community. This is just a matter of wording, not really important. Otherwise, the concept is clearly explained and well-documented in previous publications.

We agree that Schelkunoff equivalence principle is a more accurate term, and the paper has been revised accordingly in section 2. Thanks for pointing this out.

3. Results

In the paragraph *Radiating Unit Cell with Closed Bandgap*, the authors present the unit cell employed to achieve accidental degeneracy of the eigenmodes at the Gamma point. From the results, it is clear that the unit cell achieves the desired behavior. However, this unit cell is different from that in [23], [51] and [52]. Is the design solely based on the optimization process described in the Methods section or have the authors built on transmission line theory as in [23] or used another theoretical starting point?

The design is based solely on the optimization process described in the Methods section. This has been elaborated more in the paper in the paragraph starting line 568.

In the paragraph *Analytical Beam-Pointing Directions*, the authors use ϵ_e in (1a) to (2b). However, ϵ_r was used for the formulae in [38]. Could the authors explain this difference? In my opinion, it would be useful to motivate the definition of ϵ_e in page 8.

In the paper, the quantity ϵ_e refers to the effective relative permittivity of the individual unit cells. For the structure shown in [38], the unit cells are constructed from subwavelength square-shaped slots, which do not disturb the dispersion relation of the unit cells (compared to a parallel plate waveguide). Hence, the effective relative permittivity of the unit cells therein is merely that of the underlying dielectric medium (ϵ_r). However, for the proposed PEX unit cell in Fig. 2a, the perturbations to the unit cells are significant, and the dispersion relation is quite different from that of an unperturbed parallel plate waveguide. Thus, an effective relative permittivity (ϵ_e) is required to be defined, and it can be approximated from ($\epsilon_e = \lambda_o^2 / \lambda_{PEX}^2$), where λ_o is the free space wavelength, and λ_{PEX} is the wavelength inside the PEX cavity. At the Γ -point, λ_{PEX} is equal to the periodicity of the unit cells, i.e. the size of the unit cells d . This discussion has been added to the paper in page 9.

Fig. 3b shows the achievable beam pointing direction. However, it is not clear to me the range of ψ angles used to obtain the lines for the different modes. If I have understood correctly, the empty regions in the visible region in Fig. 3b correspond to directions where pointing cannot be achieved. Does this mean that for electrically larger antennas (narrower beams) the portion of the visible region covered by the generated beams will be reduced?

To obtain the different beam-pointing directions in Fig. 3b, all the possible plane wave directions (ψ) have been considered ($\psi \in [0, 2\pi]$). Indeed, the empty portions of the visible region in Fig. 3b correspond to directions where the generated beam cannot be pointed (at the Γ -point). Thus, for extremely large antennas with very narrow beams, the portion of the visible region that can be practically covered by

the generated beams will be reduced, because of the narrower beamwidth. This discussion has been added to the paper in lines **232** and **239**.

From this discussion it is clear that the achieved scan range from the proposed PEX antenna is limited, as it is only possible to generate pencil beams along some predefined paths. Nevertheless, it is possible to extend the scan range further in the future by incorporating additional scanning techniques, such as placing the entire PEX array on a simple rotating pedestal that can be mechanically rotated. This will enable the full coverage of the upper hemisphere, where the elevation of the pencil beam is controlled by the PEX antenna, and a corresponding azimuth is set by the PEX antenna and fine tuned by the mechanical motor. Additionally, there are alternative techniques that can be envisioned to extend the scan range such as leveraging technologies that can change the effective permittivity of the medium filling the PEX cavity. Notably, the entire permittivity of the medium filling the PEX cavity is required to be changed in unison, i.e. the individual unit cells are not required to be individually tuned. One possibility to achieve this is by replacing the dielectric substrate with voltage-controlled materials such as ferroelectric materials [61], and controlling the permittivity of the material by DC biasing the ground and top plates of the PEX cavity. Another possibility is to fill the PEX cavity with an artificial dielectric such as a bed of nails going through holes in the ground plane, and mechanically controlling the effective permittivity by changing the height of nails going through the holes. These techniques and others can be implemented to extend the scan range of the PEX antenna to achieve full 2D spatial coverage, and the proposed PEX array implementation is just the beginning of a new technology with many future possibilities. This discussion has been added to the paper in lines **508**, **558** and **562**.

In the paragraph *PEX array simulation*, the authors report a high simulated aperture efficiency of 84.6% for a $10\lambda \times 10\lambda$ aperture. This means that the obtained directivity is quite close to the ideal one of approximately 31 dB. However, the realized gain is only 25.1 dB due to the relatively large (44%) amount of power delivered from Side A to Side C. In other words, the almost uniform aperture distribution is obtained at the expense of coupling power to the opposite side. The authors explain that “a bigger PEX array will exhibit less power lost to the resistive terminations along side C.” In my opinion, they should also explain that with the uniform (as opposed to amplitude-tapered) radiating structure, the reduction of power lost to resistive terminations in larger PEX arrays will lead to a reduction of the aperture efficiency. I believe that this limitation should be mentioned in the introduction too, since it has an impact on the practical application of this concept.

The proposed PEX antenna array, at its heart, is a uniform leaky-wave antenna, thus, it experiences an inevitable tradeoff between aperture efficiency, radiation efficiency, and termination efficiency. The latter refers to any remaining unradiated power that is lost to the resistive terminations. In Table 1, a breakdown of the different portions of power that reach the 4 sides of the PEX array when side A is excited are shown, together with the corresponding aperture and radiation efficiencies. This power breakdown does not include the effects of the feeding network that will be later used to excite the PEX array in the experiment. For the broadside case, it is observed that 44.4% of the incident power at side A reaches the opposite side C. Note that a larger PEX array will exhibit more radiation from the leaky wave, and less power will reach side C as a result, however, the aperture efficiency will inevitably drop as well. To mitigate this tradeoff, it is possible to design a PEX antenna array with tapered radiation from the unit cells, which is a standard practice for leaky-wave antennas that aim at achieving high aperture and radiation efficiencies simultaneously. This discussion has been added to the paper in lines **278** and **298**.

In Table 1, could the authors report for each case the excited side and the power coupled to the other sides? In my opinion, this information can be of interest for the reader.

We added this information to Table 1, and it is introduced in the paper in line **298**.

Fig. 4 shows 3D radiation patterns for different multi-beam configurations. I believe that color maps in the UV plane would be clearer. If the authors choose to leave the figure as is, please include at least the color bar to read the values. Also, which magnitude are the authors plotting? Please explain. The same comments apply to Fig. 7.

The magnitude plotted in Figs. 4,7 (and Fig. 14 in the supplementary information) corresponds to the normalized far-field radiation patterns in linear scale. This has been explained in the captions of the corresponding figures. Additionally, color maps in the UV plane of the same magnitudes have been added to Figs. 4,7 (and Fig. 15 in the supplementary information) with a color bar to read the values.

The paragraph *PEX Array Experiment* reports measurements carried out by a planar near-field scanner from NSI. I am not an expert on antenna measurement. However, I do not understand why it is challenging to plot the E- and H-plane. Using the fields sampled on the plane by the near-field probe, one can apply a near- to far-field transformation to obtain the far-fields at the desired θ - ϕ pairs. Could the authors explain me with more detail? I am just curious, as long as the compared planes are the same, the authors can keep the preferred reference system.

An illustration of a pencil beam generated at an arbitrary angle and plotted using the Theta-Phi (θ - ϕ) coordinate system has been included Fig. 17a. In addition, the same descriptive pencil beam is included in Fig. 17b using the Az-over-EI (Az/EI) coordinate system. This shows how both spherical coordinate systems are defined, and how they compare.

It is possible to apply a series of coordinate system transformations and rotations to the measured results, and extract the E-plane and H-plane radiation patterns from them, however, this would involve many mathematical calculations, inevitably adding numerical errors to the measured results. Thus, it was preferred to plot the measured results directly using their native (Az/EI) coordinate system, given that both (θ - ϕ) and (Az/EI) spherical coordinate systems are identical along the cardinal planes (x-z and y-z planes). In particular, for the x-z plane ($\phi = 0^\circ$) the elevation angle (θ) is identical to Az, whereas for the y-z plane ($\phi = 90^\circ$) the elevation angle (θ) is identical to EI. Thus, both coordinate systems can be used interchangeably along or around the cardinal planes. Since, the proposed PEX antenna mostly scans the generated pencil beams close to these cardinal planes, the measured and simulated results can be compared directly even though they are technically plotted using different coordinate systems.

We still realize that having both coordinate systems can be confusing, especially that they use similar words to describe different things. To prevent any confusion, the words azimuth and elevation are exclusively used in the paper now to describe quantities related to the spherical coordinate system (θ - ϕ). Whereas, for the Az-over-EI coordinate system the abbreviated letters “Az” and “EI” are used to describe the corresponding quantities.

The previous discussion and changes have been added to the paper in lines **259, 262, 290, 372, 649, 652** and **Fig. 6**, and the supplementary document in lines **99, 142, 156, 158** and **Figs. 9-12, Fig. 17** and **Table 1**.

In line 248, please replace “patters” with “patterns.”

Thank you for pointing out this typo, the paper has been revised accordingly.

In Fig. 6, could the authors comment on the difference between simulated and measured patterns? In Figs. 6b-c, the measured patterns for high elevation angles are more directive than the ones predicted by full-wave simulations, whereas the measured directivities for zero elevation are lower than the predicted ones. Also, in line 258, I think the figure shows a “good agreement” rather than a “great agreement.”

Indeed, there is a discrepancy between the measured and simulated peak gain values in Fig. 6, even though the results correspond to a PEX antenna with the same physical size. This discrepancy can be explained by observing the phase-shifters’ transmission response in supplementary section S1, where it is clear that the transmission amplitude changes as a function of the achieved phase shift. Hence, unlike the full-wave simulations, the fabricated PEX antenna is excited with a progressively phase-shifted excitation with various amplitudes. This causes a discrepancy between the measured and simulated results in Fig. 6. The maximum discrepancy between the measured and simulated peak gain when side A is excited is around 2.22dB and when side B is excited is around 3.13dB. Needless to say that a better optimized phase shifter design with a more constant transmission amplitude for different phase shifts would allow a closer agreement between the measured and simulated peak gain values. This discussion has been added to the paper in line **421**.

In line 282, is there any particular reason for choosing 13 GHz as center frequency? Do the authors have a particular application in mind?

For the proof-of-concept demonstration in the paper, the frequency of operation (13.1GHz) has been chosen for demonstration purposes. It is a good compromise as it is high enough for the resulting 2D array design to be physically small and manageable, and it is low enough to allow the use of inexpensive commercial connectors, cables and terminations. A comment has been added to the paper in line **221**.

Also in line 282, I think that further detail should be included in this paragraph about the bandwidth of the proposed array. In the supplementary material (supplementary Figs. 6-7), the authors show a quite stable beam between 12.5 GHz and 13.7 GHz, is this the achievable bandwidth? I suggest to choose a criteria (1 or 3 dB directivity drop) to give some tentative numbers for the relative bandwidth.

The proposed PEX antenna array achieves radiation by exciting a travelling leaky-wave mode, hence, the generated pencil beams undergo a slight spatial scanning as the frequency of operation is changed. The 1dB and 3dB bandwidths achieved by the generated pencil beams at the center frequency (13.1GHz) for different plane-wave directions (ψ) have been added to Table 1. For example, at broadside, the generated pencil beam achieves a 2.5% 1dB-bandwidth and a 4.5% 3dB-bandwidth. Note that this bandwidth limitation is caused by the spatial scanning of the generated pencil beams, and not beam degradation. Being a LWA, the proposed PEX array can in principle be operated over a wider bandwidth while maintaining a directive beam, without beam degradation (as shown in supplementary section S2). This discussion has been added to the paper in line **306**.

Furthermore, the measured pencil beams exhibit an average of 2.1% 1dB-bandwidth and 7.8% 3dB-bandwidth when side A is excited, and an average of 3.3% 1dB-bandwidth and 8.2% 3dB bandwidth when

side B is excited. It is observed that the measured bandwidths are slightly wider than the simulated values reported earlier in Table 1, even though the phase shifters used in the experiment have a frequency-dependent phase shift (see supplementary section S1), whereas the full-wave simulated PEX array is excited with a frequency invariant progressive phase shift. This can be partially justified by the fact that the measured PEX array is 1/4 of the size of the full-wave simulated version, hence, the measured pencil beams exhibit a much wider beamwidth, which can lead to a slightly wider directivity bandwidths. On the other hand, although the fabricated PEX antenna is symmetric, there is a very small discrepancy between the measured directivity bandwidth for the generated pencil beams from sides A and B. This is caused by the different dispersion characteristics of the phase shifters used to excite sides A and B, as well as small unwanted bounces and reflections around the measurement setup which may differ for the two sides. This discussion has been added to the paper in line **436**.

References

In the references section, [46] duplicates [38]

Thank you for pointing this out, the duplication has been removed from the revised paper.

Concluding remarks

The authors present the experimental validation for the Peripherally Excited (PEX) array concept. This concept has been recently presented by the authors in some of their previous works [38-43]. Some preliminary steps towards the experimental demonstration had been already taken in [41] and [43] without considering the radiating part of the system. Here, the authors build on their previous works [51]-[52] to design this radiating part and the Huygens sources modifying the structure in [38], Fig. 10. I agree with the authors that the reported experiments have not been shown before, but I suggest to present with more clarity the evolution with respect to previous developments. In other words, the novel contributions and their impact should be highlighted.

We agree that a more detailed breakdown of the previous steps that are relevant to the proposed design is beneficial for the paper. The following sentences have been added to the introduction in lines **92**, **118** and **127**.

The PEX antenna is based on the Huygens' Box structure [48-52], and is excited by Huygens' sources situated along the periphery of a radiating aperture according to the Huygens' equivalence principle (described below). Notably, the Huygens' box concept has been experimentally validated previously [48-51], where it was shown that it is possible to excite travelling plane waves inside a fully-closed metallic cavity, using peripherally-placed Huygens' sources. This is a quite remarkable feat as metallic cavities only inherently support standing waves, and the PEX array concept leverages these travelling plane waves and uses them to excite a radiating leaky-wave structure.

The Huygens' box enabled the excitation of arbitrary plane waves within a closed cavity. In the PEX antenna concept, the Huygens' box is made leaky by appropriately perforating its top plate, and exciting it with the underlying electronically-steered plane waves. This paper herein introduces the first experimental demonstration of the PEX antenna concept, while highlighting several new features and capabilities of this new antenna concept.

In the proposed implementation, a Huygens' source is constructed from a coaxial feed backed by an array of metallic vias, which is entirely compatible with standard PCB fabrication, and can be easily embedded inside commercial two-layer dielectric substrates. Furthermore, the proposed PEX unit cell is capable of generating directive pencil beams at broadside and tilted radiation directions, by adopting techniques in closing the bandgap for broadside leaky-wave antennas [53,54].

On the other hand, although the authors report the limitations of the system in terms of the achievable pointing directions, other important limitations are somehow overlooked. For instance, the trade-off between power coupled to the opposite sides and the aperture efficiency should be discussed. Likewise, some guidelines about the achievable bandwidth should be introduced in the paper. These are important aspects that can hinder the application of the proposed antenna system for MIMO and automotive radar applications and the impact of the proposed concept on the field.

We agree with these comments, the paper has been revised accordingly to include discussions about these limitations in lines 278 and 298 for the coupled-power aperture-efficiency tradeoff, and 306 and 436 for the bandwidth discussion, as discussed in the previous comments.

Otherwise, I think that the paper is complete for understanding and technically correct. Except for the points mentioned above, the theoretical and experimental results of the paper accompanied by sufficient analyses, discussions and justifications.

We thank the reviewer for the positive disposition on our paper.

Reviewer #3:

We have carefully considered every comment in the Reviewers' reports. We believe that these comments helped to improve our manuscript, and we would like to thank the reviewers for that. We would also like to thank the reviewers for the timely handling of our manuscript.

The paper presents the practical implementation and experimental validation of peripherally excited array antenna, a concept previously introduced by the same authors. It basically consists in an array of slots excited by a slow wave propagating in the parallel plate waveguide formed by the slot plate and the ground plane. The slow wave is excited by sources located at the periphery of the radiating aperture, and its linear phase front is rotated by properly selecting the phase of the sources. Radiation is achieved through the interaction of the slow wave with the periodic perturbation introduced by the slots, according to a leaky wave mechanism, for which the radiating beam direction is determined by the relative orientation of the slow wave wavefront and the periodicity axes. Consequently, the radiated beam is scanned along a certain 1D path in the uv plane when the orientation of the wave front of the exciting wave is changed.

Note that the proposed PEX antenna is symmetric with respect to the x and y axes, which enables the generation of pencil beams along multiple orthogonal scanned paths as demonstrated in Fig. 3b. This has been further clarified in the paper in lines 237 and 497.

The paper is clear and well written, the proposed development is interesting, and it contains sufficient novelty to deserve publication. However, some aspects of the proposed solution should be better clarified, and references to similar solutions should be added. In particular, the scanning mechanism of the proposed solution is the same of the so-called "series fed continuous transverse stub", or CTS, in which the rotation of the phase front is usually achieved through mechanical rotation. More recent works have proposed the use of switching sources exciting a rotationally symmetric planar lens. A few relevant references are suggested below:

[C1] W. Milroy, "The continuous transverse stub (CTS) array: basic theory, experiment, and application," in 1991 Antenna Applications Symposium, 1991.

• <https://www.thinkom.com/technology/>

[C2] K. Tekkouk, J. Hirokawa, R. Sauleau, and M. Ando, "Wideband and large coverage continuous beam steering antenna in the 60-GHz band," IEEE Trans. Antennas Propag., vol. 65, no. 9, pp. 4418–4426, 2017.

[D1] Y. B. Li, R. Y. Wu, W. Wu, C. B. Shi, Q. Cheng, and T. J. Cui, "Dual physics manipulation of electromagnetic waves by system-level design of metasurfaces to reach extreme control of radiation beams," Adv. Mater. Technol., vol. 2, no. 1, p. 1600196, 2017.

[D2] J. Ruiz-García, E. Martini, C. D. Giovampaola, D. González-Ovejero and S. Maci, "Reflecting Luneburg Lenses," early access in IEEE Transactions on Antennas and Propagation.

We thank the reviewer for the positive disposition on our paper.

Indeed, continuous transverse stub arrays and switched-beam antennas are relevant to our proposed design. The following has been added to the introduction starting from lines **48, 64, 88, 91, 100, 237, 289** and **489**.

An interesting alternative antenna concept to phased arrays is the “Continuous Transverse Stubs (CTS) Array” [C1]. This concept does not require phase shifters, instead, it typically relies on some relative mechanical rotation between its constituent components [C2]. The CTS array usually comprises a parallel-plate waveguide perforated at the top with a 1D array of long slots, which are excited with a plane wave orthogonal to the long side of the slots. The slots radiate a pencil beam in free space through a travelling leaky-wave antenna (LWA) operation. The generated pencil beam can be scanned in both elevation and azimuth along predefined paths, by mechanically changing the relative orientation between the incident plane wave and the 1D slots. In [C2], this CTS array concept has been realized using a fully-metallic structure with three separate metal layers. These metal layers are implemented using contactless gap waveguides, allowing the mechanical rotation of the top radiating layer relative to the bottom feeding layers, thus changing the relative orientation between the excited plane wave and the 1D slots. This leads to a continuous scanning of the generated pencil beam in both azimuth and elevation along a single predefined path. Note that the 1D slots can only be excited from a single side, which limits the possible scan planes from the structure. To achieve full-space scanning, it is required to mechanically rotate the three metal layers together, in addition to an independent relative rotation between them.

On the other hand, numerous switched-beam antenna designs have been proposed [A,B,D1,D2] that are operated in a similar manner to the CTS array, and are able to combat the aforementioned tradeoff between scannability and cost. These antenna designs also eliminate the need for phase shifters, and are able to change the relative orientation between the excited plane wave and the radiating structure by using single-layer [A,D1] or multi-layer [B,D2] switched multi-port beamforming networks (lenses). These networks exhibit multiple switched-input ports situated along the focal plane/region of a lens such as a parabolic or Luneburg lens, and they leverage the Fourier-transform properties of the lens to excite a LWA structure with switched plane waves, by activating the different elementary input sources. Thus, this LWA structure is operated using a switched-beam operation, and each input port corresponds to a discrete pencil beam in space situated along predefined paths. However, one difficulty with these switched-beam antennas is that they cannot scan the generated pencil beams continuously, which is the price of the switched-beam operation, and usually the 3-dB beamwidth of the generated beams needs to be carefully optimized for an overlapping spatial coverage. This becomes more challenging when very narrow beams are required. Another potential drawback is that some lens designs such as parabolic reflectors are inherently constraint to reduced spatial scanning (angular coverage), due to eventual aberrations and spillover loss from feed elements displaced farther from the focal point. Also, these switched-beam antennas are typically not designed for broadside operation, which is inconvenient in some applications. More importantly, the LWA unit cells in [A,B,D1,D2] are only excited from along a single side of the LWA, and are not excited from the orthogonal side (although it is possible in principle), which limits the number of possible scan planes from these switched-beam antennas. In any case, a LWA unit cell that is optimized for broadside operation, while allowing excitation from the two orthogonal sides of the LWA is highly desirable, and can in principle double the number of possible scan planes from the LWA, effectively constituting multiple antennas in one shared radiating aperture.

In a nutshell, the aforementioned CTS and switched-beam antennas either require mechanical rotation or input-port switching to change the relative orientation between the excited plane wave and the slot

array. In contrast, for the proposed PEX antenna design, the plane waves are synthesized directly by individual Huygens' sources with an electronically-controlled progressive phase shift that can be continuously tuned, and the sources are situated along the periphery of the radiating aperture according to the Huygens' equivalence principle. Thus, the proposed PEX unit cell is capable of generating electronically-steered pencil beams at and around broadside. Also, the symmetric nature of the unit cell enables the excitation of the PEX array from all its sides, enabling the generation of plane waves at arbitrary orientations inside the PEX cavity (the plane wave can be excited from any direction along any angle), which leads to more possible scanned planes from the radiating aperture (at least double the scanned planes of typical CTS structures). This capability can also be leveraged to generate multiple pencil beams from the PEX array simultaneously.

It is important to highlight that the PEX array can scan the generated beam in azimuth and elevation similar to the aforementioned CTS and switched-beam antennas. However, there are numerous differences between both concepts, namely, the PEX array is capable of: (a) steering the plane-waves electronically without requiring any relative mechanical rotation between its plates or any cumbersome lenses (b) continuous spatial scanning as opposed to switched beams (c) exciting plane waves from all its sides leading to more scanned planes (d) generating a single or multiple independently-steered pencil beams. It is also worth highlighting that it is possible to leverage the peripheral sources, by specifying suitable weights, to electronically tune the side-lobe levels of the generated pencil beams, and achieve some form of beamforming. This possibility will be investigated in the future.

Using this mechanism, full coverage of the upper hemisphere can be only achieved by adding a mechanical rotation of the radiating aperture. Although this is mentioned in the paper in the section entitled "analytical beam pointing direction", such a limitation should be more clearly explained already in the Introduction. In fact, this is the price to be paid for the complexity reduction in the feeding network: controlling only the phases of the elements at the periphery of the aperture instead of the phases throughout the aperture implies that the beam can only be scanned along a curved path instead of covering a conical region.

Indeed, the achieved scan range from the proposed PEX antenna is limited, as it is only possible to generate pencil beams along some predefined paths. Nevertheless, it is possible to extend the scan range further in the future by incorporating additional scanning techniques, such as placing the entire PEX array on a simple rotating pedestal that can be mechanically rotated. This will enable the full coverage of the upper hemisphere, where the elevation of the pencil beam is controlled by the PEX antenna, and a corresponding azimuth is set by the PEX antenna and fine tuned by the mechanical motor. Additionally, there are alternative techniques that can be envisioned to extend the scan range such as leveraging technologies that can change the effective permittivity of the medium filling the PEX cavity. Notably, the entire permittivity of the medium filling the PEX cavity is required to be changed in unison, i.e. the individual unit cells are not required to be individually tuned. One possibility to achieve this is by replacing the dielectric substrate with voltage-controlled materials such as ferroelectric materials [61], and controlling the permittivity of the material by DC biasing the ground and top plates of the PEX cavity. Another possibility is to fill the PEX cavity with an artificial dielectric such as a bed of nails going through holes in the ground plane, and mechanically controlling the effective permittivity by changing the height of nails going through the holes. These techniques and others can be implemented to extend the scan range of the PEX antenna to achieve full 2D spatial coverage, and the proposed PEX array

implementation is just the beginning of a new technology with many future possibilities. This discussion has been added to the paper in lines **508**, **558** and **562**.

As a matter of fact, the restricted achievable coverage appears to be a severe limitation for the application of the proposed solution in MIMO. Also, the angles reported in eq. 2, and the presentation in Fig. 3b, which shows the angles relevant to all the possible Floquet modes, are misleading, since they fail to account for the unimodal condition. The practically usable modes for the claimed objective to obtain a high directivity pencil beam are only the ones that can be individually excited, as the ones identified by dots in Fig. 3b.

A portion of the beam-pointing directions in Fig. 3b are affected by grating lobes. A dotted black circle has been added to Fig. 3b showing the calculated critical angle for the PEX unit cell, i.e. the grating-lobe limit. All the beam-pointing angles presiding inside this circle can be achieved without the generation of grating lobes as they satisfy the unimodal condition. Thus, this critical angle constitutes a practical limit on the maximum scan range possible from the proposed PEX antenna. Needless to say that these potential grating lobes can be entirely prevented by designing a PEX unit cell with a smaller physical size d , which in turn leads to a larger value for the critical tilted angle (θ_c), and a larger dotted circle in Fig. 3b. This would allow more beam-pointing directions to satisfy the unimodal condition, and the generation of pencil beams along more of the possible beam-pointing directions predicted in Fig. 3b, without the generation of any grating lobes whatsoever. This discussion has been added to the paper in lines **234** and **392**.

Concerning the presentation of the results, some information could be added to increase clarity (some of them are included in the Supplementary material, but it would be better to have them also in the main body of the paper):

- overall efficiency of simulated and measured antennas should be indicated (for simulations, this can actually be reconstructed by elaborating the information provided for the single contributions, but it would be nice to have it directly available), and it should be specified if it includes the contribution of the feeding network.

The proposed PEX antenna array, at its heart, is a uniform leaky-wave antenna, thus, it experiences an inevitable tradeoff between aperture efficiency, radiation efficiency, and termination efficiency. The latter refers to any remaining unradiated power that is lost to the resistive terminations. In Table 1, a breakdown of the different portions of power that reach the 4 sides of the PEX array when side A is excited are shown, together with the corresponding aperture and radiation efficiencies. This power breakdown does not include the effects of the feeding network that will be later used to excite the PEX array in the experiment. For the broadside case, it is observed that 44.4% of the incident power at side A reaches the opposite side C. Note that a larger PEX array will exhibit more radiation from the leaky wave, and less power will reach side C as a result, however, the aperture efficiency will inevitably drop as well. To mitigate this tradeoff, it is possible to design a PEX antenna array with tapered radiation from the unit cells, which is a standard practice for leaky-wave antennas that aim at achieving high aperture and radiation efficiencies simultaneously. This discussion has been added to the paper in lines **278** and **298**.

On the other hand, determining the efficiency of the measured PEX array is more involved and cannot be done easily, as the fabricated PEX antenna is a multiport device (64 ports), and an accurate report of

the efficiency requires the measurement of the entire S-matrix of the antenna (64×64), which is a lengthy and tedious process. Nevertheless, the measured gain patterns are overall in good agreement with the full-wave simulation results, which strongly suggests that the practical efficiency is quite close to that reported in Table 1.

- the possible causes of the few dB discrepancies between the gain values for simulation and measurements should be discussed .

The measured gain values in Fig. 6 are significantly lower than the simulation results shown in Fig. 3. This is simply because the size of the fabricated PEX array is 1/4 that of the simulated design. Hence, the measured gain values are expected to be at least 6dB lower than the simulations. A comment has been added in the paper in line **358**.

On the other hand, there is a discrepancy between the measured and simulated peak gain values in Fig. 6, even though the results correspond to a PEX antenna with the same physical size. This discrepancy can be explained by observing the phase-shifters' transmission response in supplementary section S1, where it is clear that the transmission amplitude changes as a function of the achieved phase shift. Hence, unlike the full-wave simulations, the fabricated PEX antenna is excited with a progressively phase-shifted excitation with various amplitudes. This causes a discrepancy between the measured and simulated results in Fig. 6. The maximum discrepancy between the measured and simulated peak gain when side A is excited is around 2.22dB and when side B is excited is around 3.13dB. Needless to say that a better optimized phase shifter design with a more constant transmission amplitude for different phase shifts would allow a closer agreement between the measured and simulated peak gain values. This discussion has been added to the paper in line **421**.

- it seems that the unit cell design is only based on the real part of the propagation constant. Is the leaky wave radiation constant also considered in the design?

Yes, both the propagation constant and leakage constant were considered in the design of the PEX unit cell. To successfully operate the PEX array at and around broadside, all the relevant eigenmodes of the unit cell are required to be collocated in frequency Γ -point, and they are required to exhibit balanced Q-factors. Note that the Q-factor is directly related to the leakage constant (α), where a lower Q-factor, corresponds to a higher leakage constant, and more radiation from the individual unit cells. For the proposed PEX unit cell, the radiating cross-shaped slots are mostly responsible for achieving radiation from the unit cell, hence, their dimensions control the amount of leakage constant and Q-factor. In the design process, the dimensions of the cross-shaped slots are first determined to achieve a strong radiation from the corresponding leaky-wave mode. After that, the dimensions of the four square-shaped slots are tuned to close the bandgap at the Γ -point and enable successful radiation at and around broadside. This discussion has been added to the paper in lines **206, 210 and 575**.

- It is mentioned that the proposed antenna could be used for duplex applications with simultaneous transmit and receive operation from the same antenna. In this case, how would the coupling between sources on different sides affect the performances?

In the paper, we mention the possibility of using the proposed PEX antenna for simultaneous transmit and receive (duplex) operation, but not necessarily from the same direction. Indeed, one of the main challenges in achieving multiple-beam or duplex operation from any antenna is the mutual coupling that

may occur between the different beams, which limits the radiation efficiency of the antenna and prevents the successful duplex operation (Stein, IRE Trans. Antennas Propag., 10(5), 548-557, 1962). To minimize this unwanted mutual coupling, it is required that the transmitted and received beams exhibit a low beam-coupling factor, i.e. the beams are required to be orthogonal. The proposed PEX antenna achieves this by exhibiting orthogonal polarizations for the beams generated from sides A and B, as the polarization of the generated pencil beams is along the direction of propagation of the excited plane wave inside the PEX cavity (which is orthogonal for sides A and B). Thus, this causes a remarkably low mutual coupling between the orthogonal sides of the PEX array, and there are no coupling or degradation effects between the generated beams by sides A and B (see Table 1 and supplementary section S2), which enables the successful multiple-beam and duplex operation of the proposed PEX array. This discussion has been added to the paper in lines **116, 250, 290, 332, 388, 409, 462, 485** and **557**, and the paper now mentions duplex instead of fully duplex operation.

- Are the quantities compared in Fig. 6 consistent, in spite of the different reference systems? It is not completely clear why having the data available at all angles from near-field to far-field transformation it is not possible to get for the measurements the same field cut available from simulations.

An illustration of a pencil beam generated at an arbitrary angle and plotted using the Theta-Phi (θ - ϕ) coordinate system has been included Fig. 17a. In addition, the same descriptive pencil beam is included in Fig. 17b using the Az-over-EI (Az/EI) coordinate system. This shows how both spherical coordinate systems are defined, and how they compare.

It is possible to apply a series of coordinate system transformations and rotations to the measured results, and extract the E-plane and H-plane radiation patterns from them, however, this would involve many mathematical calculations, inevitably adding numerical errors to the measured results. Thus, it was preferred to plot the measured results directly using their native (Az/EI) coordinate system, given that both (θ - ϕ) and (Az/EI) spherical coordinate systems are identical along the cardinal planes (x-z and y-z planes). In particular, for the x-z plane ($\phi = 0^\circ$) the elevation angle (θ) is identical to Az, whereas for the y-z plane ($\phi = 90^\circ$) the elevation angle (θ) is identical to EI. Thus, both coordinate systems can be used interchangeably along or around the cardinal planes. Since, the proposed PEX antenna mostly scans the generated pencil beams close to these cardinal planes, the measured and simulated results can be compared directly even though they are technically plotted using different coordinate systems.

We still realize that having both coordinate systems can be confusing, especially that they use similar words to describe different things. To prevent any confusion, the words azimuth and elevation are exclusively used in the paper now to describe quantities related to the spherical coordinate system (θ - ϕ). Whereas, for the Az-over-EI coordinate system the abbreviated letters "Az" and "EI" are used to describe the corresponding quantities.

The previous discussion and changes have been added to the paper in lines **259, 262, 290, 372, 649, 652** and **Fig. 6**, and the supplementary document in lines **99, 142, 156, 158** and **Figs. 9-12, Fig. 17** and **Table 1**.

Other minor comments:

- The following reference could be added to explain the scanning mechanism and the derivation of eqs. 1a and 1b, not so easy to understand from the paper.

A. Bhattacharyya, "Theory of Beam Scanning for Slot Array Antenna Excited by Slow Wave," in IEEE Antennas and Propagation Magazine, vol. 57, no. 2, pp. 96-103, April 2015.

This citation has been added as a reference for the full derivation of Eqs. (1)-(2).

- In the caption of Fig. 3b "crosses" should be replaced by "circles" or "dots"

The red crosses in Fig. 3b have been replaced with red dots.

- From the plots reported in the Additional material, it seems that beam scanning is more significant for measurements than for simulations. However, this could be due the different reference systems.

Indeed, it seems that the measured results in Figs. 9-12 experience more spatial scanning with frequency when compared with the full-wave simulation results in Fig. 6-7. Nevertheless, this discrepancy looks more significant than it actually is because the gain patterns in Fig. 6-7 are plotted for an angular span of $\pm 180^\circ$, but the measured results are only plotted for an angular span of $\pm 70^\circ$, which amplifies the spatial scanning of the measured results. In any case, the measured PEX array experiences additional spatial scanning as it is excited with phase shifters that exhibit a slight dispersive amplitude/phase transmission response with frequency. Whereas, the simulated PEX array is fed with a constant progressively phase-shifted excitation without any amplitude/phase dispersive effects. This comment has been added in the supplementary material in line 104.

- In section 4 is stated that "the PEX antenna array is able to scan the generated beams along predefined azimuthal planes". In this context, what is actually meant by "azimuthal"?

To avoid any confusion, the statement in section 4 has been changed to "predefined contours".

- Throughout the paper there are some repetitions, due to the replication of some pieces of information in the Results and Method sections.

We tried to minimize repetitions without compromising the clarity of the paper.

REVIEWER COMMENTS

Reviewer #1 (Remarks to the Author):

The authors have addressed all of the comments of this reviewer. Based on the responses, I recommend that the paper is published as is.

Reviewer #2 (Remarks to the Author):

I am Reviewer #2 in the reply letter. I would like to thank the authors for taking into account my comments in the revised version of the manuscript and for the detailed reply letter.

Most of my previous comments have been satisfactorily addressed. However, I still have a few remarks about the new content. In these additional comments below, the figures, lines and references numbers correspond to those in the revised version.

In order to address my first comment (page 4 of the response letter), the authors have completed the introduction with references to CTS antennas and quasi-optical or lens beamformers. They have described some of the main features of these systems, but I respectfully disagree with some of the authors' statements.

- CTS antennas can be used in series (travelling wave leaky wave operation as in [39]) and also in parallel configuration. In the latter case, they can only scan in one plane (a mechanical rotation is still needed), but they provide excellent H-plane scanning and, more importantly, very large bandwidths. The parallel solution is extensively used for SatCom-on-the-move commercial systems and bandwidth-wise they are difficult to beat.

- Second, the authors mention "cumbersome lenses" (line 500 in the revised paper). I would like to point out that the quasi-optical and lens beam-formers in [42] and [43], respectively, are placed underneath the antenna aperture and, just as the PEX array, they save real state. Although these solutions employ two layers, the bottom layers are quite thin [42]-[43]. Hence, I would avoid talking about cumbersome solution in these cases, especially considering the additional PCB, circuits and cables needed by the proposed PEX array implementation (supplementary Figure 1).

- Third, the authors have written "exciting plane waves from all its sides leading to more scanned planes" (line 500). However, the solution in [43] also seems to provide plane waves in a discrete number of phi angles from 0 to 360°. Please clarify.

My first comment in page 6 of the response letter has been replied. Although it isn't crucial, I meant that the need of phase shifters may hinder the use of this approach in the sub-THz range, i.e., above ~100 GHz. The authors could complete their reply by providing the higher frequency they think they can target using the same commercial components.

I thank the authors for their detailed reply to the last comment in page 7 of the reply letter (continued in page 8). In summary, at the end of the day, a mechanical rotation or other sort of reconfiguration will be needed in order to provide a performance similar to that of phased arrays. Although this limitation is now discussed in section 4, I suggest the authors to state it also in the introduction to convey the advantages and disadvantages from the beginning of the document.

I have a final comment about the sentence in line 313, added as reply to the last comment in page 10. Although the beam behavior described in this sentence is true, I suggest to point out that such behavior is not desirable for communications (application described in the introduction). The instantaneous bandwidth will be given by the frequencies where the beam scanning with frequency leads to a -1 or -3 dB attenuation, which is smaller than the leaky wave antenna bandwidth. Moreover, the larger the directivity, the narrower the available bandwidth.

I suggest that the authors address the comments above in the final version.

Reviewer #3 (Remarks to the Author):

I am generally satisfied with the authors' replies to my comments, and I believe the paper is now more clear and balanced.

I just have a few residual comments:

- As a comment to CTS solution, in the revised manuscript it is stated "Note that the 1D slots can only be excited from a single side, which limits the possible scan planes from the structure" and later on, referring to both CTS and switched beam antennas, "the LWA unit cells in [40-43] are only excited from along a single side of the LWA, and are not excited from the orthogonal side" These sentences is not very clear to me since I don't see any limitation in the excitation direction that can be obtained through mechanical rotation or source switching. In case, the differences I see reside in the 1D nature of the radiating structure, as compared to the 2D periodicity of the architecture proposed in this paper, and on the intrinsic single polarization capability of the long slot-based solution. A similar consideration holds for the sentence "exciting plane waves from all its sides leading to more scanned planes" describing the difference with respect to other existing solutions, a few paragraphs later.

- In the same paragraph, among the differences between the proposed solution and the ones presented in the literature we find this one "the PEX array is capable of steering the plane-waves electronically without requiring any relative mechanical rotation between its plates or any cumbersome lenses". It is clear that, for certain applications, electronically steering can be preferable with respect to mechanical steering, however, it is less clear what the authors mean by "cumbersome" lenses (the realization of the feeding network for the proposed PEX array could be considered more cumbersome in many respects)

- In Table 1, "App. Eff." Should be probably be "Ap. Eff."

Reviewer #1:

The authors have addressed all of the comments of this reviewer. Based on the responses, I recommend that the paper is published as is.

We thank the reviewer for the positive disposition on our paper.

Reviewer #2:

We have carefully considered every comment in the Reviewers' reports. We believe that these comments helped to improve our manuscript, and we would like to thank the reviewers for that. We would also like to thank the reviewers for the timely handling of our manuscript.

I am Reviewer #2 in the reply letter. I would like to thank the authors for taking into account my comments in the revised version of the manuscript and for the detailed reply letter.

Most of my previous comments have been satisfactorily addressed. However, I still have a few remarks about the new content. In these additional comments below, the figures, lines and references numbers correspond to those in the revised version.

We thank the reviewer for the positive disposition on our paper.

In order to address my first comment (page 4 of the response letter), the authors have completed the introduction with references to CTS antennas and quasi-optical or lens beamformers. They have described some of the main features of these systems, but I respectfully disagree with some of the authors' statements.

- CTS antennas can be used in series (travelling wave leaky wave operation as in [39]) and also in parallel configuration. In the latter case, they can only scan in one plane (a mechanical rotation is still needed), but they provide excellent H-plane scanning and, more importantly, very large bandwidths. The parallel solution is extensively used for SatCom-on-the-move commercial systems and bandwidth-wise they are difficult to beat.

We agree with the reviewer's statement, we have added the following comment to the manuscript in line 64. "It is worth highlighting that unlike the travelling-wave (series-fed) operation in [39], it is also possible to operate the CTS array in a parallel-fed fashion, which offers excellent H-plane scanning and very large bandwidths [40]. This parallel CTS antenna solution is used extensively in SatComm-on-the-move commercial systems, even though mechanical rotation is still needed to achieve full-space scanning."

- Second, the authors mention “cumbersome lenses” (line 500 in the revised paper). I would like to point out that the quasi-optical and lens beam-formers in [42] and [43], respectively, are placed underneath the antenna aperture and, just as the PEX array, they save real state. Although these solutions employ two layers, the bottom layers are quite thin [42]-[43]. Hence, I would avoid talking about cumbersome solution in these cases, especially considering the additional PCB, circuits and cables needed by the proposed PEX array implementation (supplementary Figure 1).

The term “cumbersome” lenses was referring to general lens antennas that are bulky [Ref. 1] not [42],[43]. We removed 'cumbersome' from the manuscript to avoid any confusion (line 511).

Ref. 1: D.G. Bodnar, "Lens Antennas", In "Antenna Engineering Handbook", 4th-Edition, McGraw Hill, 2007, Edited by John L. Volakis.

- Third, the authors have written “exciting plane waves from all its sides leading to more scanned planes” (line 500). However, the solution in [43] also seems to provide plane waves in a discrete number of phi angles from 0 to 360°. Please clarify.

We agree in general that the design proposed in [44] ([43] in previous manuscript version) could be capable of providing switched beams in certain 0-360deg directions. To avoid ambiguity, we have made the statement in line 513 to be specific to [41,43] ([40,42] in previous manuscript version).

My first comment in page 6 of the response letter has been replied. Although it isn't crucial, I meant that the need of phase shifters may hinder the use of this approach in the sub-THz range, i.e., above ~100 GHz. The authors could complete their reply by providing the higher frequency they think they can target using the same commercial components.

We agree, expanding the frequency of operation to sub-THz range (100GHz and beyond) could present challenges, when using the current PEX array implementation with phase shifters. Thus, successful PEX array operation can only be envisioned up to the mm-wave regime, if the phase shifters are not replaced with other higher-frequency suitable alternatives. This comment has been added to the manuscript in line 552.

I thank the authors for their detailed reply to the last comment in page 7 of the reply letter (continued in page 8). In summary, at the end of the day, a mechanical rotation or other sort of reconfiguration will be needed in order to provide a performance similar to that of phased arrays. Although this limitation is now discussed in section 4, I suggest the authors to state it also in the introduction to convey the advantages and disadvantages from the beginning of the document.

A comment has been added to the introduction in line 148. “In the Discussion section, we conclude the paper with some observations and remarks, including how to extend the scan range of the PEX array, beyond its predefined scanned-planes, by a mechanical rotation or employing tunable substrates.”

I have a final comment about the sentence in line 313, added as reply to the last comment in page 10. Although the beam behavior described in this sentence is true, I suggest to point out that such behavior is not desirable for communications (application described in the introduction). The instantaneous bandwidth will be given by the frequencies where the beam scanning with frequency leads to a -1 or -3

dB attenuation, which is smaller than the leaky wave antenna bandwidth. Moreover, the larger the directivity, the narrower the available bandwidth.

The following comment has been added to the manuscript in line **322**. “This instantaneous bandwidth constitutes the actual useful bandwidth that can be used for communications, as it represents the frequency bandwidth where the beam scanning with frequency leads to a 1 or 3 dB attenuation.”

I suggest that the authors address the comments above in the final version.

Reviewer #3:

We have carefully considered every comment in the Reviewers’ reports. We believe that these comments helped to improve our manuscript, and we would like to thank the reviewers for that. We would also like to thank the reviewers for the timely handling of our manuscript.

I am generally satisfied with the authors’ replies to my comments, and I believe the paper is now more clear and balanced.

We thank the reviewer for the positive disposition on our paper.

I just have a few residual comments:

- As a comment to CTS solution, in the revised manuscript it is stated “Note that the 1D slots can only be excited from a single side, which limits the possible scan planes from the structure” and later on, referring to both CTS and switched beam antennas, “the LWA unit cells in [40-43] are only excited from along a single side of the LWA, and are not excited from the orthogonal side” These sentences is not very clear to me since I don’t see any limitation in the excitation direction that can be obtained through mechanical rotation or source switching. In case, the differences I see reside in the 1D nature of the radiating structure, as compared to the 2D periodicity of the architecture proposed in this paper, and on the intrinsic single polarization capability of the long slot-based solution. A similar consideration holds for the sentence “exciting plane waves from all its sides leading to more scanned planes” describing the difference with respect to other existing solutions, a few paragraphs later.

Yes, the difference lies in the radiating unit cell used. The proposed PEX unit cell has been designed and optimized with 2D plane-wave excitation in mind, hence, it can be operated from all four sides. However, the long-slot radiating unit cell cannot achieve radiation when excited with a plane-wave along its long edge, thus, it cannot be operated from all its four sides. This comment has been added to the manuscript in lines **61, 118**.

- In the same paragraph, among the differences between the proposed solution and the ones presented in the literature we find this one “the PEX array is capable of steering the plane-waves electronically without requiring any relative mechanical rotation between its plates or any cumbersome lenses”. It is clear that, for certain applications, electronically steering can be preferable with respect to mechanical

steering, however, it is less clear what the authors mean by “cumbersome” lenses (the realization of the feeding network for the proposed PEX array could be considered more cumbersome in many respects)

The term “cumbersome” lenses was referring to general lens antennas that are bulky [Ref. 1] not to [42],[43]. We removed 'cumbersome' from the manuscript to avoid any confusion (line **511**).

Ref. 1: D.G. Bodnar, "Lens Antennas", In "Antenna Engineering Handbook", 4th-Edition, McGraw Hill, 2007, Edited by John L. Volakis.

- In Table 1, “App. Eff.” Should be probably be “Ap. Eff.”

Thank you for pointing this out, we revised the table header accordingly.

REVIEWERS' COMMENTS

Reviewer #2 (Remarks to the Author):

The authors have addressed all my last minor comments in the previous review round. In my opinion, the paper can now be recommended for publication.